# Low-Dimension-to-High-Dimension Generalization And Its Implications for Length Generalization

**Yang Chen** [1]  **Long Yang** [1]  **Yitao Liang** [2 3]  **Zhouchen Lin** [1 2 4]

## Abstract

Low-Dimension-to-High-Dimension (LDHD) generalization, a subset of Out-of-Distribution (OOD) generalization, involves training on a low-dimensional subspace and testing in a high-dimensional space. Assuming instances are generated from latent variables reflecting problem scale, LDHD generalization captures the inherent scaling challenge of length generalization. We theoretically show that LDHD generalization is unattainable without appropriate inductive bias. Focusing on Boolean functions, we demonstrate that different architectures trained with (S)GD converge to *min-degree interpolators w.r.t. different linearly independent sets*, achieving LDHD generalization only when the target function aligns with this bias. From the perspective of LDHD generalization for length generalization, we explain the success of CoT in restructuring latent space for improved LDHD generalization. We further propose a principle for designing position embeddings to address both LDHD generalization and data format nuisances separately. Following the principle, we introduce RPE-Square, a novel embedding that enhances RPE to better handle data formats.

## 1. Introduction

Learning to reason has gained significant popularity in the machine learning community due to its impressive performance in reasoning tasks such as natural language processing (OpenAI, 2023a;b), mathematics (Frieder et al., 2023; Jelassi et al., 2023), coding (Zhang et al., 2022a), symbolic logic (Abbe et al., 2023; Garcez et al., 2022), and planning (Zhao et al., 2023; Valmeekam et al., 2023). One of the most significant challenges in learning to reason is *length generalization* (Anil et al., 2022; Zhang et al., 2022b), where models trained on small-scale instances must generalize to large-scale instances. Length generalization is crucial because the size of sample spaces often increases exponentially with the complexity of reasoning problems, leading to intractable sample complexity and computational costs for models that do not achieve length generalization.

Numerous works have investigated various techniques for length generalization, including modifications to model architectures (Shaw et al., 2018; Jelassi et al., 2023; Kazemnejad et al., 2024), transformations in data formats (Lee et al., 2023; Zhou et al., 2023), prompt engineering for Large Language Models (LLMs) (Wei et al., 2022; Feng et al., 2024). Although some of the above techniques work uniformly well across a wide class of problems, many are fragile and even ad-hoc, applicable only to specific problems with certain formats (Zhou et al., 2024). This is due to the mismatch between the inherent problem scale and the length of the input string. We illustrate it in the next Example 1.

**Example 1.** *Consider the addition learning problem where the input is a string. For an $N$-digit-plus-$N$-digit (written as $N$-addition) addition of two numbers $x$ and $y$, where $x = x_{N-1} \ldots x_0$, $y = y_{N-1} \ldots y_0$, $x_i, y_i \in \{0, \ldots, 9\}$ ($x_{N-1} > 0$ or $y_{N-1} > 0$), consider two formats: the Aligned and Reverse Format (ARF), where the instance is represented as "$\mathrm{x}_0 \ldots \mathrm{x}_{n-1} + \mathrm{y}_0 \ldots \mathrm{y}_{n-1} =$"; and the Unaligned and Reverse Format (URF), where the instance is represented as "$\mathrm{x}_0 \ldots \mathrm{x}_{n_x-1} + \mathrm{y}_0 \ldots \mathrm{y}_{n_y-1} =$", $n_c = \arg\max_i\{c_i \neq 0\}, c \in \{x, y\}$. However, the length of the input strings in neither format faithfully reflects the problem scale: in ARF, the length of the input strings is always invariant; in URF, a shorter string may correspond to a larger scale than a longer string (e.g., $s_1 =$ "$1 + 1234 =$" is $4$-addition, $s_2 =$ "$123 + 123 =$" is $3$-addition; $s_1$ is of larger scale than $s_2$ but $\mathrm{length}(s_1) < \mathrm{length}(s_2)$).*

Example 1 demonstrates the sensitivity of length generalization to the data format. Developing robust and transferable methods for scalable models requires a formulation that captures the inherent scaling challenge of length generalization and is invariant to the data format nuisance.

[1]State Key Lab of General Artificial Intelligence, School of Intelligence Science and Technology, Peking University [2]Institute for Artificial Intelligence, Peking University [3]Beijing Institute for General Artificial Intelligence [4]Pazhou Laboratory (Huangpu), Guangzhou, Guangdong, China. Correspondence to: Zhouchen Lin <zlin@pku.edu.cn>, Yitao Liang <yitaol@pku.edu.cn>.

*Proceedings of the $42^{nd}$ International Conference on Machine Learning*, Vancouver, Canada. PMLR 267, 2025. Copyright 2025 by the author(s).

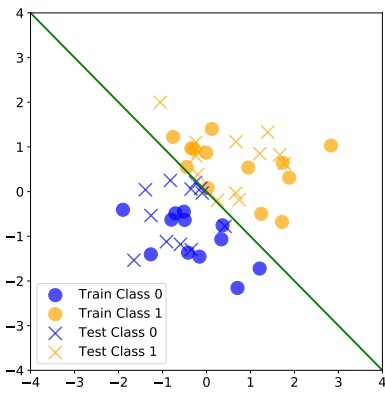 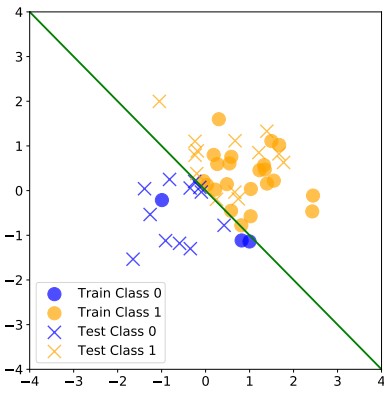 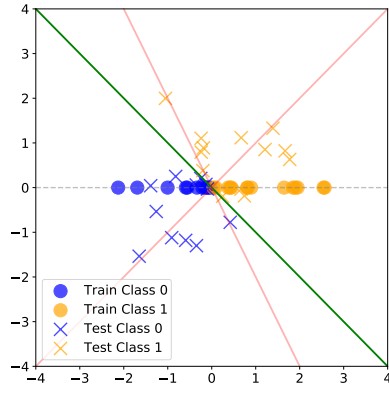

(a) In-Distribution Generalization.  (b) (Typical) OOD Generalization.  (c) LDHD Generalization.

*Figure 1.* Illustrative comparison of in-distribution generalization, typical OOD generalization, and LDHD generalization. (a) In-distribution generalization assumes identical training and testing distributions. (b) Typical OOD generalization involves a shift between training and testing distributions, which remain relatively "close" (e.g., sharing support or having small distributional distances). (c) LDHD generalization features a training distribution restricted to a low-dimensional subspace and a testing distribution on a high-dimensional space, often vastly different. While LDHD is a type of OOD generalization, its structured shift poses unique challenges, as training data provide no clues about the additional dimensions' contribution to the label.

## 1.1. Our Main Works

**Low-Dimension-to-High-Dimension Generalization Perspective for Length Generalization.** We propose Abstraction 1 that disentangles the problem scale and the data format, which provides a more precise formulation for the analysis of length generalization.

**Abstraction 1** (Data Generation in Length Generalization). The data generation process of an instance of scale $n$ with the concept $c$ is as follows:

1. A latent variable $h$ is sampled from a subspace $\Sigma^n$ of dimension $n$;

2. The label $y$ is determined by the concept $c$ and the latent variable $h$, i.e., $y = c(h)$;

3. The latent variable $h$ is transformed to an input sequence by the data format mapping $\phi$.

In Abstraction 1, the scale of the instance is captured by the dimension of the latent space from which the instance is sampled. The scale shift in length generalization is characterized by Low-Dimension-to-High-Dimension (LDHD) generalization in the latent space. In other words, the generalization from "short" instances to "long" instances corresponds to the generalization from the low-dimension latent subspace of the training data to the high-dimension latent subspace. The computation of the task is represented by the concept $c$, which is defined on the latent space and independent of the data format. The data format decides how a hidden variable is mapped to an actual input sequence. The next Example 2 illustrates the abstraction in the addition.

**Example 2** (Addition). *Let $\Sigma = \{0, \ldots, 9\}^2$. The instance*

*of $n$-addition $x_{n-1} \ldots x_0 + y_{n-1} \ldots y_0$ can be represented by the latent vector $h_n = [(x_0, y_0), \ldots, (x_{n-1}, y_{n-1})]$. The $k$-th dimension of $h_n$ corresponds to the $(k-1)$-th digits of the two addenda (i.e., $x_{k-1}$ and $y_{k-1}$). The expansion of the state vector in the dimension corresponds to the increase in the addendum digits. In the length generalization of the addition task, the scale shift from $N_0$-addition to $N$-addition can be seen as a generalization from the low-dimensional latent space $\Sigma^{N_0}$ to the high-dimensional latent space $\Sigma^N$. The label, i.e., the sum of the addenda, is determined by the computation of the addition task. The input string is determined by both the latent variable and the data format (e.g., ARF or URF).*

Example 2 demonstrates that each instance of $n$-addition is generated from an $n$-dimension latent variable via a data format mapping. Length generalization of addition requires addressing both LDHD generalization in the latent space, which captures the computation of addition as the scale increases, and the data format, which may hinder the model in learning the addition operator. In general, from the LDHD generalization perspective, length generalization involves two key aspects: LDHD generalization in the latent space and the nuisance of the data format. LDHD generalization reflects the scale shift between the training and test data, which is the inherent challenge of length generalization.

**No-Free-Lunch Theorem of LDHD Generalization.** The main challenge of LDHD generalization is that the testing space has extra dimensions compared to the training space. As a result, the testing space contains instances with orthogonal components to the training space. The training samples cannot reveal any information about how these components

contribute to the results. For instance, in Example 2, learning with $N_0$-addition solely does not tell how $(N_0 + 1)$-th digits to $N$-th digits contribute to the result unless we have the prior knowledge that the addition of each digit shares the same process. Figure 1 shows an intuitive illustration for the challenge of LDHD generalization of linear models on $\mathbb{R}^2$, compared to in-distribution generalization and typical OOD generalization settings. When learning a linear decision boundary from the low-dimension training data, we need to know the slope of the decision boundary as a prior; otherwise, we will fail to achieve LDHD generalization: While all the three solid lines in Figure 1c perfectly separate the low-dimension training data, only the true decision boundary can achieve LDHD generalization.

We formalize the above challenge as *No-Free-Lunch Theorem (Wolpert & Macready, 1997; Wolpert, 2002) of LDHD Generalization*. Theorem 1 shows that no algorithm could achieve LDHD generalization uniformly for all tasks, which necessitates the use of prior knowledge in the learning process in order to achieve LDHD generalization.

**Inductive Bias of Models and Algorithms for LDHD Generalization.** In practice, the prior knowledge is usually incorporated via the inductive bias of the learning algorithms and the models. To develop LDHD generalizable models, we investigate the inductive bias of different architectures trained by (S)GD. While random feature models are shown to be *min-degree interpolators* (Abbe et al., 2023), Theorem 2 shows that random feature models with projections, where inputs are transformed by projections before fed to the random feature models, converge to *min-degree interpolators w.r.t. different linearly independent sets*. Furthermore, we consider Position-Only Linear Attentions with Advice (PLAA), i.e., linear attentions whose attention scores are determined only by positions, with additional hints about the scales of the instances. This model can be seen as a simplified abstraction of decoder-only transformers with a special focus on positional relations that are considered crucial for length generalization. Theorems Theorem 3 and Theorem 4 show that PLAA with Absolute Position Embedding (APE) and PLAA with Relative Position Embedding (RPE) converge to min-degree interpolators w.r.t. different linearly independent sets. The results illustrate the limitation of PLAA with APE in LDHD generalization and the potential of PLAA with RPE to overcome the limitation.

**Implications of the LDHD Generalization Perspective for Practical Length Generalization Techniques.** The LDHD generalization perspective further provides insights into practical length generalization techniques. In Section 5.1, we discuss the role of Chain-of-Thought (CoT) (Wei et al., 2022) in length generalization. We show that CoT can be seen as a change of the hidden space, where each dimension of the latent space is augmented with an additional middle state. This transformation could facilitate LDHD generalization in the latent space for some problems. Additionally, in Section 5.2, we propose a principle of position embedding design for length generalization with Transformers: We need to handle both the inherent LDHD generalization and the nuisances such as the data format in the design of the position embeddings. Following this principle, we propose a novel position embedding named RPE-Square. Our experiments show that the RPE-Square evidently enhances the RPE with the ability to handle the nuisance of the unaligned data format.

The next sections of the paper are organized as follows: In Section 3, we introduce the formal definition and No-Free-Lunch Theorem of LDHD generalization; in Section 4, we present the results of the inductive bias of different models trained by gradient and how the inductive bias influences LDHD generalization; in Section 5, we show how the LDHD perspective helps to understand and design practical techniques for length generalization.

### 1.2. Notations

We use $[n]$ to represent the set of numbers $\{1, \ldots, N\}$. We denote the set of all functions from the set $\mathcal{X}$ to the set $\mathcal{Y}$ as $\mathcal{F}_{\mathcal{X}, \mathcal{Y}}$. We define $\mathrm{Proj}(x, V)$ as the coordinate of the projection of $x$ onto the space spanned by $V$ with the basis $V = [v_1, \ldots, v_r]$, i.e., $[\mathrm{Proj}(x, V)]_i = \langle x, v_i \rangle$ for all $i = 1, \ldots, r$. We use $A^*$ to denote the Kleene closure of the set $A$, i.e., $A^* = \bigcup_{k=0}^{\infty} A^k$. We use $\deg(p)$ to denote the degree of the polynomial $p$. We represent the set of $N \times N$ upper triangular matrix as $\mathcal{U}_N$.

## 2. Related Work

**Length generalization in reasoning problems.** Length generalization is a key challenge in learning to reason, typically interpreted as the ability to learn with small-scale instances of a task and generalize to unseen large instances of the same task (Anil et al., 2022; Zhou et al., 2023). Various reasoning tasks are considered to investigate length generalization, including arithmetic (Jelassi et al., 2023; Feng et al., 2024), Boolean logic (Abbe et al., 2023; d'Ascoli et al., 2023), symbolic reasoning (Zhang et al., 2022b), etc. Despite the rich literature on length generalization on specific reason tasks, few works have considered challenges and overconditions of length generalization for general problems. The existing works that analyze general length generalization mainly focus on the change of the input sequence length (Xiao & Liu, 2023; Ahuja & Mansouri, 2024; Huang et al., 2024). This formulation conflates the inherent scaling challenges with the effects of data format, making it difficult to precisely capture the challenges of length generalization. To address this, our work proposes disentangling length generalization into two distinct aspects: LDHD generalization

in the latent space, which characterizes the challenges associated with increasing scale, and the data format nuisance.

**Inductive Bias of Model Architectures and Algorithms.** In the situation of learning with overparatermization or incomplete information, proper inductive bias is essential to select the true model from the hypothesis (Neyshabur, 2017; Bartlett et al., 2021; Teney et al., 2024). One of the most studied scenarios is the inductive bias of different model architectures under (S)GD. Previous research shows (S)GD combined with different model architectures lead to different effects of implicit regularization, such as different norms of the learnable parameters Gunasekar et al. (2017); Bartlett et al. (2021) and different complexity characterizations of the models (Razin & Cohen, 2020; Razin et al., 2021; Abbe et al., 2023). We specially mention Abbe et al. (2023), which proposes that models trained with (S)GD are biased towards min-degree profile interpolators in the context of Boolean functions, which do not achieve general LDHD generalization. Our results show that the min-degree profile bias does not hold for all models. We further show that models with different architecture can converge to min-degree profile interpolators under different linearly independent sets when trained with (S)GD. This partially explains how model architectures can affect length generalization in reasoning.

## 3. LDHD Generalization

**Definition 1** (Low-Dimension-to-High-Dimension Generalization). Suppose that $\mathcal{X}$ is a sample space and $\mathcal{X}_1, \mathcal{X}_2$ are two subspaces such that $\mathcal{X}_1 \subset \mathcal{X}_2 \subset \mathcal{X}$ and $\dim(\mathcal{X}_1) < \dim(\mathcal{X}_2)$. Consider a concept class $\mathcal{C} \subset \mathcal{F}_{\mathcal{X},\mathcal{Y}}$, two distributions $\mathcal{D}_1, \mathcal{D}_2$ where $\text{supp}(\mathcal{D}_1) = \mathcal{X}_1$ and $\text{supp}(\mathcal{D}_2) = \mathcal{X}_2$, and a learning algorithm $\mathcal{A} : (\mathcal{X} \times \mathcal{Y})^* \mapsto \mathcal{F}_{\mathcal{X},\mathcal{Y}}$. We say *low-dimension-to-high-dimension generalization* of the concept class $\mathcal{C}$ from $\mathcal{D}_1$ to $\mathcal{D}_2$ is achieved by the algorithm $\mathcal{A}$ with $m$ samples and $\epsilon$ error if

$$\mathbb{E}_{X^m \sim \mathcal{D}_1^m, X_{m+1} \sim \mathcal{D}_2} \left[ \ell \left( \hat{f}_{X^m, c}(X_{m+1}), c(X_{m+1}) \right) \right] \leq \epsilon,$$

where $\hat{f}_{X^m, c}$ is the function learned by the algorithm $\mathcal{A}$ from the training samples $X^m$ labeled by the concept $c$ and $\ell : \mathcal{Y} \times \mathcal{Y} \mapsto \mathbb{R}$ is the loss function.

Definition 1 extends the Independent Identical Distribution (IID) assumption in PAC learning theory and considers a special shift between the training data and the testing data. This shift is particularly challenging because the testing space is of strictly higher dimension than the training space. Generally, it is impossible to fully capture the structure of the training space from the testing data as the training data reveal no information on how components in the orthogonal subspace contribute to the output. Therefore, there is no algorithm that can always guarantee to learn the concept from the training data. Theorem 1 formally states the nonex-

istence of universal algorithms for LDHD generalization.

**Theorem 1** (No-Free-Lunch Theorem of LDHD Generalization). *Suppose that the two sets $\mathcal{X}$ and $\mathcal{Y}$ are finite. For some $N > N_0$, consider two subsets $\mathcal{X}_{N_0}$, $\mathcal{X}_N$ of $\mathcal{X}$ such that $\mathcal{X}_{N_0} \subsetneq \mathcal{X}_N \subseteq \mathcal{X}$ and $\dim(\mathcal{X}_{N_0}) = N_0 < N = \dim(\mathcal{X}_N)$. Let $c_1, c_2 \in \mathcal{F}(:= \mathcal{F}_{\mathcal{X}_N, \mathcal{Y}})$ be two concepts such that $c_1(x) = c_2(x)$ for all $x \in \mathcal{X}_{N_0}$. For any $c \in \mathcal{F}$ and $\mathcal{X}' \subseteq \mathcal{X}$, define $\mathcal{F}/(c \mid \mathcal{X}') := \{f \in \mathcal{F} \mid f(x) = c(x) \text{ for all } x \in \mathcal{X}'\}$. Let $\ell : \mathcal{Y} \times \mathcal{Y} \mapsto \mathbb{R}$ be the loss function. For any distribution $\mathcal{D}(\mathcal{X}_N)$ such that $\text{supp}(\mathcal{D}(\mathcal{X}_N)) = \mathcal{X}_N$:*

$$\sum_{f \in \mathcal{F}/\left(c_1 \mid \mathcal{X}_{N_0}\right)} \mathbb{E}_{x \sim \mathcal{D}(\mathcal{X}_N)} \left[ \ell\left(c_1(x), f(x)\right) \right]$$

$$= \sum_{f \in \mathcal{F}/\left(c_2 \mid \mathcal{X}_{N_0}\right)} \mathbb{E}_{x \sim \mathcal{D}(\mathcal{X}_N)} \left[ \ell\left(c_2(x), f(x)\right) \right].$$

Theorem 1 necessitates the consideration of structural assumptions on the concept class such that a learning algorithm could identify the target concept from the hypothesis with the imperfect information provided by the low-dimensional training data. For example, the concept class of linear classifiers with fixed weight vector on $\mathcal{X} = \mathbb{R}^d$, i.e., $\mathcal{C} = \{\text{sgn}\left(w_0^{\mathsf{T}} x + b\right) \mid w_0 \in \mathbb{R}^d, b \in \mathbb{R}\}$ with the $d'$-dimensional training sample space $\mathcal{X}_1 = \mathbb{R}^{d'} \times \{0\}^{d-d'}$ and the $d$-dimensional testing sample space where $d' < d$. For any concept $c \in \mathcal{C}$, a learning algorithm could not identify the true concept $c$ solely from the training data from $\mathcal{X}_1$ and the hypothesis of all linear classifiers. However, if a learning algorithm exploits the structure of the concept class, i.e., by fixing the weight vector to $w_0$ in prior, it can compute the target bias from the training data in $\mathcal{X}_1$ and then identify the target concept $c$.

We further investigate how model structures can affect LDHD generalization. We show in Section 4 that different models trained with (S)GD can be seen as min-degree interpolators under different functional bases in the context of Boolean functions, which is a joint inductive bias of the model structures and (S)GD. This insight suggests a principle for model design to achieve LDHD generalization: ensure the concept class is "low-degree" under the linearly independent set induced by the model structure.

LDHD generalization captures the challenge of scale shift in length generalization. Intuitively, length generalization means that a model trained with small-scale instances of a reasoning problem can perform well on large-scale instances of the problem. To formalize the intuition, we consider a typical instance being generated from a hidden variable $h$ that represents the "core" of the instance and is transformed to the model input by a data format mapping. The hidden space is of the form $\Sigma^*$ for some domain $\Sigma$. The dimension

$n$ of the subspace from which the hidden variable is sampled, $n = \arg\max_k\{h \in \Sigma^k, h \notin \Sigma^{k+1}\}$, reflects the increase in the scale of the problem. Besides, we define the concept class on the hidden space, depicting that the change of the data format does not change the computation of the concepts. Abstraction 1 describes the pipeline of the data generation in length generalization.

*Remark* 1. Some works observe length generalization on certain tasks without domain-specific priors, using only a small fraction of large-scale data during pretraining. This does not contradict our No-Free-Lunch Theorem, as "no" and "few" large-scale examples are fundamentally different. Prior work (Jelassi et al., 2023) shows that exposure to a tiny fraction of long sequences in pre-training, which is called *priming*, can significantly improve length generalization. From our perspective, this is because the presence of long sequences, even in small quantities, prevents the model from learning an overly simple interpolator that fails to extrapolate. For example, in the addition task, training only on 3-digit numbers may lead the model to ignore digits beyond the third, while including just a few 5-digit examples compels it to process later digits, enhancing extrapolation.

## 4. Main Results

We show theoretically how different models succeed or fail to achieve LDHD generalization as the effect of the inductive bias of the architectures under (S)GD. We focus on Boolean functions. More specifically, in the context of Boolean functions, we have $\Sigma = \{\pm 1\}$, $\mathcal{X} = \Sigma^N$, $\mathcal{X}_n = \Sigma^n \times \{1\}^{N-n}$ for $n = 1, \ldots, N$. The set of all Boolean functions potentially considered is $\mathcal{F} = \mathcal{F}_{\mathcal{X},\mathbb{R}}$. Our analysis can be naturally extended to tasks over finite alphabets by considering their binary representations.

We consider LDHD generalization from $N_0$ to $N$ for some $N_0 < N$. Define $I(f)$ as the minimal set $I$ of indices that the function $f$ can be represented as a function of $x_I$, i.e., $I(f) := \arg\min_{I \subset [N]} |I|$ such that $f(x) = \tilde{f}(x_I)$ for some function $\tilde{f}$ and all $x \in \mathcal{X}$. A function $f$ is $k$-sparse if $|I(f)| \le k$.

Before presenting the theoretical results, we introduce two concepts *degree profile w.r.t. linearly independent set* and *min-degree interpolator w.r.t. linearly independent set*, which extend the concept *degree profile* and the concept *min-degree interpolator*, respectively. We use the two concepts to characterize the inductive bias of different model architectures under (S)GD.

**Definition 2** (Degree Profile w.r.t. Linearly Independent Set $\mathcal{B}$). Suppose that $\mathcal{B} = \{b_1, \ldots, b_R\}$ is a linearly independent set of functions in $\mathcal{F}$ and $D = \max_{b \in \mathcal{B}} \deg(b)$. Let $f \in \mathcal{F}$ be a function in the subspace spanned by $\mathcal{B}$, i.e., $f = \sum_{i=1}^{R} \hat{f}_{\mathcal{B}}(b_i) b_i$ for some $\hat{f}_{\mathcal{B}}(b_i) \in \mathbb{R}, i = 1, \ldots, R$. The *de-*

*gree profile* of the function w.r.t. $\mathcal{B}$, denoted by $\mathrm{DegP}_{\mathcal{B}}(f)$, is a $(D+1)$-tuple where $D_i = \sum_{b \in \mathcal{B}, \deg(b) = D+1-i} \hat{f}_{\mathcal{B}}(b)^2$ for $i = 1, \ldots, D+1$. The order of degree profiles is identical to the lexicographic order of the corresponding $D$-tuples.

**Definition 3** (Min-Degree Interpolator w.r.t. Linearly Independent Set $\mathcal{B}$). Suppose that $\mathcal{B} = \{b_1, \ldots, b_R\}$ is a linearly independent set of functions in $\mathcal{F}$. Let $\mathcal{X}'$ be a subset of the sample space $\mathcal{X} = \{\pm 1\}^N$. Denote the set of all interpolators on $\mathcal{X}'$ for the concept $c$ by $\mathcal{G}_{\mathcal{X}',c}$, i.e., $\mathcal{G}_{\mathcal{X}',c} = \{g \in \mathcal{F} \mid g(x) = c(x) \text{ for all } x \in \mathcal{X}'\}$. A function $g$ is called the min-degree interpolator w.r.t. $\mathcal{B}$ on $\mathcal{X}_0$ for the concept $c$ if $g \in \mathcal{G}_{\mathcal{X}',c}$ and $\mathrm{DegP}(g) \le \mathrm{DegP}(g')$ for all $g' \in \mathcal{G}_{\mathcal{X}',c}$.

### 4.1. Random Feature Model with Projection

We first consider the random feature model (RFM) and a class of its variants, i.e., Random Feature Models with Projection (RFMP; see Definition 4). RFM is widely employed as approximations of practical neural network models in theoretical studies. By comparing the inductive biases introduced by RFM and RFMP under various projections, we demonstrate the importance of incorporating prior knowledge to achieve LDHD generalization and this prior knowledge can be effectively embedded through model design.

**Definition 4** (Random Feature Model with Projection). Suppose that $V = [v_1, \ldots, v_r] \in \mathbb{R}^{N \times r}$ satisfies $V^\mathsf{T} V = I_r$. A random feature model with projection w.r.t. $V$ is

$$f_{\mathrm{RFMP}}^{V,K}(x; a) = \frac{1}{\sqrt{K}} \sum_{k=1}^{K} a_k \sigma\left(\langle w_k, \mathrm{Proj}\,(x, V)\rangle + b_k\right),$$

where $K$ is the number of random features, $\sigma$ is the activation function, $a = [a_1, \ldots, a_K]^\mathsf{T}$ is the learnable parameter, and $w_k \sim \mathcal{N}(0, I_r/r)$, $b_k \sim \mathcal{N}(0, 1/r)$ for all $k = 1, \ldots, K$.

The original RFM can be seen as a special instance of RFMP with $V = I_N$. Technically, we follow the strongly expressive condition (Abbe et al., 2023) for the activation function $\sigma$. Abbe et al. (2023) shows that the RFM converges to the min-degree interpolator when initialized at 0 and trained with GD. However, this is not the case for all RFMP models. We show in Theorem 2 that an RFMP model converges to a min-degree interpolator w.r.t. a linearly independent set determined by the set $V$.

**Theorem 2.** *Suppose that* $V = [v_1, \ldots, v_r] \in \mathbb{R}^{N \times r}$ *satisfies* $V^\mathsf{T} V = I_r$. *Define the set* $\mathcal{B}(V)$ *of independent functions as*

$$\mathcal{B}(V) := \left\{\chi_T^V(x)\right\}_{T \subseteq [r]},$$

*where*

$$\chi_T^V(x) = \prod_{t \in T} \sum_{n=1}^{N} (v_t)_n x_n.$$

Let $a_t$ be the learnable parameter at the timestep $t$ in the training process where the learnable parameter $a$ is initialized at $a_0 = 0$ and optimized with gradient descent/gradient flow under $\ell_2$ loss on $\mathcal{X}_{N_0}$. Let $\mathcal{G}_{N_0,c,V}$ be the set of all interpolators on $\mathcal{X}_{N_0}$ for the concept $c^*(x) = c(\mathrm{Proj}(x, V))$ that is $O_N(1)$-sparse. Then we have, as $K \to \infty, t \to \infty$,

$$f_{RFMP}^{V,K}(x; a_t) \to \arg \min_{g \in \mathcal{G}_{N_0,c,V}} \mathrm{DegP}_{\mathcal{B}(V)}(g).$$

When $V = I_N$, the linearly independent set $\mathcal{B}(V)$ is the Fourier basis of the Boolean functions, and Theorem 2 implies that the RFM converges to the min-degree interpolator. From Theorem 2, we see that an RFMP model with the projection matrix $V$ can achieve LDHD generalization only if the target concept coincides with the min-degree interpolator w.r.t. $\mathcal{B}(V)$. Specially, for the RFM, we have:

**Corollary 1.** *For any $f \in \mathcal{F}$ such that $I(f) \not\subset [N_0]$, the min-degree interpolator does not achieve LDHD generalization from $\mathcal{X}_{N_0}$ to $\mathcal{X}_N$ and thus the RFM initialized at 0 and trained with GD does not achieve LDHD generalization from $\mathcal{X}_{N_0}$ to $\mathcal{X}_N$.*

Corollary 1 shows that the min-degree interpolator and thus the RFM model can only achieve LDHD generalization for a very restricted set of functions that are only dependent on $x_{[N_0]}$. Achieving LDHD generalization with RFMP models requires prior knowledge of the concept class to design the projection. Example 3 illustrates how LDHD generalization is possible for the target function with dependence beyond $x_{[N_0]}$ by choosing a proper projection.

**Example 3.** *Consider the target function $f(x) = 4x_1 + 3x_2$, $N_0 = 1$, and $N = 2$. The min-degree interpolator on $\mathcal{X}_{N_0}$ is $f_1(x) = 4x_1$, which does not achieve LDHD generalization on $\mathcal{X}_N$. In the RFMP model, if we choose*

$$V = \begin{bmatrix} 0.8 & 0.6 \\ 0.6 & -0.8 \end{bmatrix},$$

*then we have*

$$\mathcal{B}(V) = \{1, 0.8x_1 + 0.6x_2, 0.6x_1 - 0.8x_2,$$
$$(0.8x_1 + 0.6x_2)(0.6x_1 - 0.8x_2)\}.$$

*The min-degree interpolator w.r.t. the linearly independent set $\mathcal{B}(V)$ on $\mathcal{X}_{N_0}$ is $f_2(x) = 4x_1 + 3x_2 = f(x)$, which achieves LDHD generalization on $\mathcal{X}_N$.*

### 4.2. Position-Only Linear Attention with Advice

In this subsection, we investigate Position-Only Linear Attention with Advice (PLAA), which can be seen as a simplification of decoder-only Transformers (Definition 5), with a special focus position embeddings that are considered pivot to the length generalization of the Transformers (Shaw et al., 2018; Jelassi et al., 2023).

**Definition 5** (PLAA). Define the advice function $n : \mathcal{X} \mapsto \{0, \ldots, N\}$ such that $n(x) = \arg \max_n \{x_n = -1\}$ if there exists $k \in [N]$ such that $x_k = -1$ and $n(x) = 0$ otherwise. We additionally define $e_0 := 0$. A PLAA model is

$$f_{\mathrm{PLAA}}(x; A) = x^\intercal A e_{n(x)},$$

where $A \in \mathcal{U}_N$ is the learnable parameter and $e_n$ denotes the vector with a 1 in the $n$-th coordinate and 0's elsewhere.

We further elaborate on the intuition behind the PLAA models. In the generation process of a decoder-only (linear) Transformer with position embeddings given input $\mathtt{s} = \mathtt{s_1} \ldots \mathtt{s_n}$, the attention is computed by the query at the position $n$ and the keys at the positions $i \leq n$. The position of the query is special, advising the length of the input and reflecting the scale of the instance ideally. The PLAA model captures this feature and introduces the notation $n(x)$ to reflect the dimension of the subspace that $x$ belongs to. To further simplify and focus on the position embeddings, we assume that the value of each $x_i$ is identical to itself (i.e., we fix the value matrix in the attention to $I$ and thus $W_V x_i = x_i$) since the value head is not central to length generalization: the interpolator is expected to learn a suitable value head. We reasonably suppose a correct value head to focus on the specific challenge in length generalization. Additionally, we assume the attention is only related to positions, highlighting the standalone impact of position embeddings on inductive bias and length generalization. The contribution of the interaction between the position embeddings is $[e_1, \ldots, e_n]^\intercal A_{[n],[n]} e_n = A_{[n],n}$ for some upper triangle matrix $A$. When the input is embedded to length $N$ but the query is still made at the position $n$, the output of the model is $x^\intercal A e_{n(x)}$, i.e., the output of the PLAA model. Therefore, the PLAA model is a simplification of decoder-only Transformers focusing on the impact of the position embeddings on length generalization. For a more detailed elaboration on PLAA, see Appendix B.

In Definition 5, we directly parameterize the PLAA model directly with the attention matrix. In practice, however, the attention matrix is typically computed by the interaction between the position embeddings. Therefore, we consider the PLAA models with the Absolute Position Embedding (APE) and the Relative Position Embedding (RPE), respectively. See Definitions 6 and 7. Note that we consider Generalized RPE (GPRE) in Definition 7 because the RPE can be seen as a special instance of the GRPE with $\mathcal{U} = \mathcal{U}_{\mathrm{RPE}} = \{D_1, \ldots, D_N\}$, where $D_k$ is a $k$-th upper diagonal matrix such that $(D_k)_{ij}$ is 1 if $j = i + k - 1$ and 0 otherwise. We seek a more general result applicable to all similar parameterization methods to the RPE.

**Definition 6** (PLAA with APE). A PLAA model with APE is

$$f_{\mathrm{PLAA}}^{\mathrm{APE}}(x; P) = x^\intercal \left( M_N^u \circ P^\intercal P \right) e_{n(x)},$$

where $M_N^u \in \mathbb{R}^{N \times N}$ is the upper triangle mask, i.e., $(M_N^u)_{ij}$ is 1 if $i \leq j$ and 0 otherwise, and $P \in \mathbb{R}^{d_P \times N}$ is the learnable parameter of the model.

**Definition 7** (PLAA with GRPE). For $\mathcal{U} = \{U_1, \ldots, U_r\}$, a PLAA moodel with GRPE is

$$f_{\text{PLAA}}^{\text{GRPE},\mathcal{U}}(x; p) = x^\intercal \left( \sum_{i=1}^r U_i p_i \right) e_{n(x)},$$

where $U_i \in \mathcal{U}_N, i = 1, \ldots, r$ are upper triangle matrices that satisfy $\langle U_i, U_i \rangle = 1$ for all $i = 1, \ldots, r$ and $(U_i)_{kl} (U_j)_{kl} = 0$ for all $i \neq j$ and $1 \leq k, l \leq N$, and $p = [p_1, \ldots, p_r]^\intercal \in \mathbb{R}^r$ is the learnable parameter.

*Remark* 2. The condition $(U_i)_{kl} (U_j)_{kl} = 0$ for all $i \neq j$ and $1 \leq k, l \leq N$ in Definition 7 means each element in the position-only attention is characterized by at most one parameter. This condition generalizes the property of RPE that $A_{i,j} (i \leq j)$ is only parameterized by $p_{j-i}$.

For the PLAA model with APE, Theorem 3 shows that it converges to the min-degree interpolator w.r.t. the linearly independent set $\mathcal{B}_N^{\text{PLAA}}$.

**Theorem 3.** *Define the set $\mathcal{B}_N^{PLAA}$ as*

$$\mathcal{B}_N^{PLAA} := \{b_{ij}^{PLAA}(x)\}_{1 \leq i \leq j \leq N},$$

*where*

$$b_{ij}^{PLAA}(x) = \begin{cases} -\frac{1-x_j}{2} \prod_{k=j+1}^N \frac{1+x_k}{2}, & i = j, \\ x_i \frac{1-x_j}{2} \prod_{k=j+1}^N \frac{1+x_k}{2}, & i < j. \end{cases}$$

*Suppose that $d_P \geq N$. Let $P_t(\alpha)$ be the learnable parameter at the timestep $t$ in the training process where the learnable parameter $P$ is initialized at $P_0(\alpha)$ such that $P_0^\intercal(\alpha) P_0(\alpha) = \alpha I_N$ ($\alpha > 0$), and optimized with gradient descent/gradient flow under $\ell_2$ loss (denoted by $L(P)$) on $\mathcal{X}_{N_0}$. Define $P_\infty(\alpha) := \lim_{t \to \infty} P_t(\alpha)$. Let $\mathcal{G}_{N_0, A^*}^{PLAA}$ be the set of all interpolators on $\mathcal{X}_{N_0}$ for the concept $c(x) = f_{PLAA}(x; A^*)$. If $L((P_\infty(\alpha)) = 0$ for all $0 < \alpha \leq \alpha_0$ ($\alpha_0 > 0$ is some constant) and $\hat{P} := \lim_{\alpha \to 0} P_\infty(\alpha)$, then*

$$f_{PLAA}^{APE}(x; \hat{P}) = \arg \min_{g \in \mathcal{G}_{N_0, A^*}^{PLAA}} \text{DegP}_{\mathcal{B}_N^{PLAA}}(g).$$

For any function $f$ that can be represented by a PLAA model, there exists a matrix $A^f \in \mathbb{R}^{N_0 \times N_0}$ such that $f(x) = \sum_{1 \leq i \leq j \leq N_0} A_{ij}^f b_{ij}^{PLAA}(x)$ for all $x \in \mathcal{X}_{N_0}$. Consequently, for $f_{\text{PLAA}}^{\text{APE}}(x)(x; \hat{P}) = \sum_{1 \leq i \leq j \leq N} \hat{A}_{ij} b_{ij}^{PLAA}(x)$, we have $\hat{A}_{ij} = 0$ for all $j \geq i > N_0$. This implies that the PLAA with APE cannot achieve LDHD generalization for the concept $c(x) = f_{\text{PLAA}}(x; A^*)$ if $A_{ij}^* \neq 0$ for some $j \geq i > N_0$, which partially explains the limitation of APE for length generalization.

The PLAA with GRPE can overcome the aforementioned limitation of the PLAA with APE in length generalization.

Theorem 4 characterizes that the PLAA with GRPE converges to the interpolator that minimizes the degree-profile w.r.t. the linearly independent set $\mathcal{B}_{\text{PLAA}}^{\text{GRPE},\mathcal{U}}$.

**Theorem 4.** *For the $\mathcal{U} = \{U_1, \ldots, U_r\}$, define*

$$\mathcal{B}_{PLAA}^{GRPE,\mathcal{U}} := \left\{ \sum_{1 \leq i \leq j \leq N} (U_k)_{ij} b_{ij}^{PLAA}(x) \right\}_{1 \leq k \leq r}.$$

*Let $p_t$ be the learnable parameter at the timestep $t$ in the training process where the learnable parameter $p$ is initialized at $p_0 = 0$ and optimized with gradient descent/gradient flow under $\ell_2$ loss on $\mathcal{X}_{N_0}$. Let $\mathcal{G}_{N_0, p^*}^{GRPE,\mathcal{U}}$ be the set of all interpolators on $\mathcal{X}_{N_0}$ for the concept $c(x) = f_{PLAA}^{GRPE,\mathcal{U}}(x; p^*)$. Then we have, as $t \to \infty$,*

$$f_{PLAA}^{GRPE,\mathcal{U}}(x; p_t) \to \arg \min_{g \in \mathcal{G}_{N_0, p^*}^{GRPE,\mathcal{U}}} \text{DegP}_{\mathcal{B}_{PLAA}^{GRPE,\mathcal{U}}}(g).$$

With the inductive bias of the PLAA with GRPE, Corollary 2 states that LDHD generalization can be achieved if and only if the target concept can be represented by the elements in $\mathcal{B}_{\text{PLAA}}^{\text{GRPE},\mathcal{U}}$ that have a dependence on $\mathcal{X}_{N_0}$.

**Corollary 2.** *Consider the concept $c(x) = f_{PLAA}^{GRPE,\mathcal{U}}(x; p^*)$:*

$$\sum_{k=1}^r c_k \sum_{1 \leq i \leq j \leq N} (U_k)_{ij} b_{ij}^{PLAA}(x).$$

*Under the conditions of Theorem 4, the PLAA with GRPE achieves LDHD generalization from $\mathcal{X}_{N_0}$ to $\mathcal{X}_N$ if and only if*

$$\{k \mid (U_k)_{[N_0],[N_0]} = 0\} \subseteq \{k \mid c_k = 0\}.$$

*Remark* 3. While the projection in the RFM and the position embeddings in the PLAA may introduce an inductive bias that benefits LDHD generalization, they can reduce the expressiveness of the models. The point is to align the models with the concept class, incorporating a strong inductive bias while maintaining sufficient expressiveness.

## 5. Further Implications

We discuss further implications of the LDHD generalization perspective for practical length generalization techniques.

### 5.1. Chain-of-Thought for Length Generalization

While the Chain-of-Thought (CoT) can lead to more variety in the length of testing samples, it is widely used as an effective technique to improve the length generalization in various reasoning tasks. This seems contradictory if considered in the original space of the input sequence. However, from the LDHD generalization perspective, we can see that CoT intrinsically changes the underlying hidden space by

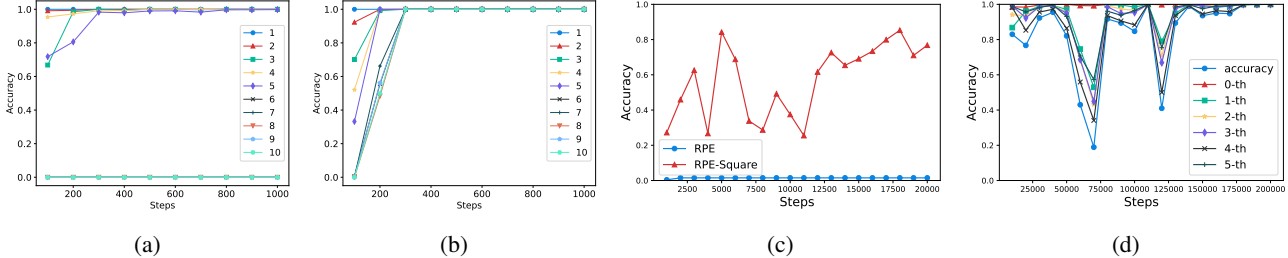

*Figure 2.* Length generalization of Transformer with RPE and RPE-Square in the unaligned copy and the URF addition tasks. **Unaligned Copy**: Transformers with RPE (a) and RPE-Square (b) are trained on lengths 1–5 for 1000 steps and tested on lengths 1–10. While both models generalize in-distribution, only RPE-Square achieves out-of-distribution generalization for lengths 6–10. **URF Addition**: Models are trained on URF 4-addition and tested on URF 5-addition. (c) Both the models are trained for 20000 steps. The comparison result shows that the RPE fails while the RPE-Square succeeds in achieving length generalization. (d) RPE-Square trained for 200,000 steps achieves nearly perfect accuracy, with digitwise accuracy shown for each $z_k$.

extending each dimension with a "middle" variable and does not lead to the dimensional increase in the hidden space. For example, consider the $n$-addition without CoT and the $n$-addition with CoT. In the case without CoT, the instance $x_{n-1} \ldots x_0 + y_{n-1} \ldots y_0 = z_n \ldots z_0$ corresponds to the latent state $h_n = [(x_0, y_0), \ldots, (x_{n-1}, y_{n-1})] \in \Sigma^n$ for $\Sigma = \{0, \ldots, 9\}^2$. In the case with CoT, one step of predicting $z_t$ corresponds to the latent state

$$h_n = [(x_0, y_0, z_0), \ldots, (x_{t-1}, y_{t-1}, z_{t-1}),$$
$$(x_t, y_t, *) \ldots, (x_{n-1}, y_{n-1}, *)] \in \bar{\Sigma}^n$$

for $\bar{\Sigma} = \Sigma \times \{*, 0, \ldots, 9\}$, where $*$ is a special element indicating undetermined values. CoT does not cause the LDHD generalization challenge but extends the domain $\Sigma$, leading to a more easily learnable target concept.

### 5.2. Position Embeddings for Length Generalization

Position embeddings are considered closely related to the length generalization in Transformers. Our analysis suggests a principle for position embedding design: consider the inherent LDHD generalization and the data format nuisance separately. To further elaborate, consider the length generalization of the URF addition with CoT. While RPE can capture the recursive structure of the addition problem and could lead to LDHD generalization in the latent space (Zhou et al., 2023; Jelassi et al., 2023), it fails to work for the URF addition with CoT, due to the nuisance of the URF.

Following the principle, we design a novel position embedding called RPE-Square to handle the nuisance of the unaligned data format. On the one hand, we keep the RPE structure for the inherent LDHD generalization. On the other hand, we deal with the unalignment by considering the distances to several special tokens (e.g., [BOS], $+$, and $=$). These considerations lead to the RPE-Square, in which we compute the relative values between the distances to special tokens. More concretely, the RPE-Square$_{i,j}$ for the

query at $j$ and the key at $i$ is

$$\sum_{1 \le k \le j, 1 \le l \le i} \frac{\exp\left((W_Q x_j)^\intercal (W_K x_l)\right)}{\sum_{1 \le l' \le j} \exp\left((W_Q x_j)^\intercal (W_K x_{l'})\right)}$$
$$\times \frac{\exp\left((W_Q x_i)^\intercal (W_K x_k)\right)}{\sum_{1 \le k' \le i} \exp\left((W_Q x_i)^\intercal (W_K x_{k'})\right)} R_{(j-l)-(i-k)},$$

where $W_Q$ and $W_K$ are the weight matrices for the query and the key, respectively. We replace RPE$_{j-i}$ with RPE-Square$_{i,j}$ in the Transformer.

RPE-Square incorporates prior knowledge of LDHD generalization and unaligned data formats by combining RPE with a mechanism to handle unaligned formats. The position embedding for a query $j$ and a key $i$ is determined by the *relative distance of relative distances*–the difference between the relative distance of $j$ to the position of some token $x_l$, and $i$ to the position of some token $x_k$, parameterized by $R_{(j-l)-(i-k)}$. This design, which inspires the name RPE-Square, uses a weighted average over all $1 \le l \le j, 1 \le k \le i$, where the weights are derived from the product of attention scores between $x_j$ and $x_l$, and $x_i$ and $x_k$. This approach enables automatic learning of special tokens and is particularly suited for tasks involving unaligned data formats, such as URF addition. Further illustration is provided in Appendix D.1.

We compare the length generalization performance of RPE and RPE-Square in two tasks: *Unaligned Copy* and *URF Addition*. In the unaligned copy, the input is a string whose length is not aligned to a fixed length, and the target is one copy of the input string. An unaligned copy instance of scale $n$ is like "[BOS] x_0 ... x_{n-1} = x_0 ... x_{n-1} [EOS]". The URF addition is illustrated in Example 1. To examine length generalization, the models are trained only on small-scale instances but evaluated on instances of larger scales. More details of the experiments are in Appendix D.2. The experiment results are presented in Figure 2. In the experiments, RPE-Square, the position embedding derived according to the LDHD generalization perspective, can achieve

length generalization in both tasks, while RPE fails. The experiment shows that RPE-Square effectively improves over RPE in handling the unalignment in the data format.

## 6. Conclusion and Discussion

We propose the LDHD generalization perspective for length generalization, which disentangles the problem into two aspects: LDHD generalization in the latent space and handling data format nuisances. From this perspective, we introduce the No-Free-Lunch Theorem of LDHD generalization, highlighting the necessity of inductive bias for achieving length generalization. Using Boolean functions, we investigate the inductive biases of different model architectures trained with (S)GD and how these inductive biases contribute to LDHD generalization. Our perspective on LDHD generalization further elucidates the role of CoT and leads to the principle of position embedding design in length generalization.

For future work, while our theory is established with simplified models, it is crucial to investigate inductive bias in more practical and complex models. Additionally, developing a paradigm for position embedding design based on the proposed principles is a valuable avenue.

## Impact Statement

This paper presents work whose goal is to advance the field of Machine Learning. There are many potential societal consequences of our work, none which we feel must be specifically highlighted here.

## Acknowledgements

Z. Lin and Y. Liang were supported by National Key R&D Program of China (2022ZD0160300). Z. Lin was also supported by the NSF China (No. 62276004). Y. Liang was additionally supported by CCF Baidu Open Fund.

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

## A. Background on Boolean Analysis

In this section, we include a brief background on Boolean analysis essential to this work. We refer to O'Donnell (2014) for further details and comprehensive coverage of Boolean analysis.

**Fourier Expansion.** A Boolean function $f : \{-1,1\}^n \mapsto \mathbb{R}$ can always be represented by

$$f(x) = \sum_{T \subset [n]} \hat{f}(T)\chi_T(x), \tag{1}$$

where $\chi_T(x) = \pi_{i \in [T]} x_i$. The polynomial in (1) is called the Fourier expansion of the Boolean function $f$. The number $\hat{f}(T)$ is the Fourier coefficient of $f$ on $T$. The set $\{\chi_T(x)\}_{T \subset [n]}$ forms a basis, named Fourier basis, for the product space w.r.t the inner product defined by

$$\langle f, g \rangle = \mathbb{E}_{x \sim U(\{-1,1\}^n)} [f(x)g(x)].$$

**Degree and Degree Profile.** The degree of a Boolean function $f$ is the degree of its Fourier expansion, which is a polynomial. The degree profile of $f$, denoted by $\mathrm{DegP}(f)$, is a $(n+1)$-tuple where $\mathrm{DegP}_i(f) = \sum_{T \subset [n], |T| = n+1-i} \hat{f}^2(T)$. The order between two degree profiles is lexicographic. The degree profile can be roughly seen as the distribution of the degrees of the monomials in the polynomial. Intuitively, the degree profile reflects the "complexity" of the Boolean function. A lower-degree-profile Boolean function uses fewer variables or combines them more simply than a higher-degree-profile one.

## B. Detailed Explanation of PLAA

In this section, we illustrate how PLAA and its variants abstract the effect of position embedding on the LDHD generalization of decoder-only Transformers.

### B.1. Construction of PLAA

A typical linear attention at query $n$ can be expressed as

$$f_{\mathrm{LA}}(x; W_Q, W_K, W_V, B) = \sum_{i \leq n} [(W_Q x_n)^{\mathsf{T}} (W_K x_i) + B_{i,n}] W_V x_i, \tag{2}$$

where $W_Q, W_K, W_V$ are the query, key, value matrices, respectively, and $B \in \mathbb{R}^{N \times N}$ is the position bias. To focus on the impact of the position embeddings, we fix $W_V = I$ and consider position-only attention score. Then we have

$$f_{\mathrm{PLA}}(x; B) = \sum_{i \leq n} B_{i,n} x_i = \sum_{i \leq n} e_i^{\mathsf{T}} B e_n x_i.$$

Rewriting $B$ as $A$ and restricting $A$ to an upper triangular matrix, we have

$$f_{\mathrm{PLA}}(x; A) = \sum_{i=1}^{n} e_i^{\mathsf{T}} A e_n x_i = x^{\mathsf{T}} A e_n, \quad \text{where } A \in \mathcal{U}_N.$$

In the computation of attention, the query position is asymmetric to other positions and can reflect the scale of the instance. To formalize the intuition, we introduce the notation $n(x) := \arg\max_{i \in [N]} \{x_i = -1\}$, which represents the last dimension where $x_i = -1$. In the LDHD generalization framework, $n(x)$ can be interpreted as the lowest dimension of the subspaces containing $x$, reflecting the dimension of $x$. We take $n(x)$ as the query position of $x$. Specifically, when $x$ is all-ones, it is treated as an "empty" sequence. In this case, we fix its output to align with the definition of PLAA. By the above derivation, we obtain Definition 5.

### B.2. Construction of PLAA with APE

The expression of the linear attention with APE is slightly different from (2). In practice, APE is commonly added to the token embeddings. Therefore, we omit the position bias and add APE to each $x_i$ when computing the attention score (we slightly abuse the notation of $x_i$ to denote the token embedding), i.e.,

$$f_{\mathrm{LA}}^{\mathrm{APE}}\left(x; W_Q, W_K, W_V, \{p_i\}_{i \in [N]}\right) = \sum_{i \leq n} [W_Q (x_n + p_n)]^{\mathsf{T}} [W_K (x_i + p_i)] W_V x_i,$$

where $p_i$ is the position embedding for the position $i$.

Similar to the derivation in Appendix B.1, we have

$$f_{\text{PLAA}}^{\text{APE}}(x; W_Q, W_K) = \sum_{i \leq n(x)} p_{n(x)}^{\intercal} W_Q^{\intercal} W_K p_i x_i = \sum_{i \leq n(x)} e_{n(x)}^{\intercal} P^{\intercal} W_Q^{\intercal} W_K P e_i x_i,$$

where $P$ is the learnable position embedding matrix such that $p_i = Pe_i$ for all $i \in [N]$. Without loss of generality, we absorb the learnable parameters $W_Q, W_K$ into $P$, obtaining

$$f_{\text{PLAA}}^{\text{APE}}(x; P) = \sum_{i \leq n} x_i^{\intercal} e_i^{\intercal} P^{\intercal} P e_{n(x)} = x^{\intercal} \left( M_N^u \circ P^{\intercal} P \right) e_{n(x)}.$$

### B.3. Construction of PLAA with GRPE

RPE can be seen as a reparameterization of the position bias matrix such that the entries on the same diagonals share learnable parameters, i.e.,

$$B = \sum_{i=1}^{N} D_i p_i,$$

where $D_k$ is the $k$-th upper diagonal matrix and $p_i$ is the learnable parameter for all $1 \leq k \leq N$.

We generalize RPE so that our model can cover more general position biases. A natural extension is to choose general upper triangular matrices, denoted by $\mathcal{U}_{\text{GRPE}} = \{U_1, \ldots, U_r\}$, instead of $\mathcal{U}_{\text{RPE}} = \{D_1, \ldots, D_N\}$. To normalize, we suppose $\langle U_i, U_i \rangle = 1$ for all $i = 1, \ldots, r$. (This condition does not hold for $\mathcal{U}_{\text{RPE}}$. We can slightly modify the matrices by replacing $D_k$ with $\frac{1}{N+1-k} D_k$ to satisfy the condition.) We further require $(U_i)_{kl}(U_j)_{kl} = 0$ for all $i \neq j$ and $1 \leq k, l \leq N$, i.e., each $B_{i,j}$ is characterized by at most one learnable parameter. Using the reparameterization

$$B = \sum_{i=1}^{r} U_i p_i$$

in the derivation of PLAA, we obtain PLAA with GRPE in Definition 7.

## C. Proofs

### C.1. Proof for Theorem 1

Let $\mathcal{D}_1 := \mathcal{D}\left(\mathcal{X}_N \mid x \in \mathcal{X}_{N_0}\right)$ be the conditional distribution given that $x \in \mathcal{X}_{N_0}$ and $\mathcal{D}_2 := \mathcal{D}\left(\mathcal{X}_N \mid x \notin \mathcal{X}_{N_0}\right)$ be the conditional distribution given that $x \notin \mathcal{X}_{N_0}$

$$\sum_{f \in \mathcal{F}/\left(c_1 \mid \mathcal{X}_{N_0}\right)} \mathbb{E}_{x \sim \mathcal{D}(\mathcal{X}_N)} \left[ \ell\left(c_1(x), f(x)\right) \right]$$

$$= P_{\mathcal{D}(\mathcal{X}_N)}(x \in \mathcal{X}_{N_0}) \sum_{f \in \mathcal{F}/\left(c_1 \mid \mathcal{X}_{N_0}\right)} \mathbb{E}_{x \sim \mathcal{D}_1} \left[ \ell\left(c_1(x), f(x)\right) \right]$$

$$+ P_{\mathcal{D}(\mathcal{X}_N)}(x \notin \mathcal{X}_{N_0}) \sum_{f \in \mathcal{F}/\left(c_1 \mid \mathcal{X}_{N_0}\right)} \mathbb{E}_{x \sim \mathcal{D}_2} \left[ \ell\left(c_1(x), f(x)\right) \right]$$

$$= P_{\mathcal{D}(\mathcal{X}_N)}(x \in \mathcal{X}_{N_0}) V_1(c_1) + P_{\mathcal{D}(\mathcal{X}_N)}(x \notin \mathcal{X}_{N_0}) V_2(c_1),$$

where

$$V_1(c) := \sum_{f \in \mathcal{F}/\left(c \mid \mathcal{X}_{N_0}\right)} \mathbb{E}_{x \sim \mathcal{D}_1} \left[ \ell\left(c(x), f(x)\right) \right],$$

$$V_2(c) := \sum_{f \in \mathcal{F}/\left(c \mid \mathcal{X}_{N_0}\right)} \mathbb{E}_{x \sim \mathcal{D}_2} \left[ \ell\left(c(x), f(x)\right) \right].$$

Similarly, we have

$$\sum_{f \in \mathcal{F}/\left(c_2|_{\mathcal{X}_{N_0}}\right)} \mathbb{E}_{x \sim \mathcal{D}(\mathcal{X}_N)}\left[\ell\left(c_2(x), f(x)\right)\right] = P_{\mathcal{D}(\mathcal{X}_N)}(x \in \mathcal{X}_{N_0})V_1(c_2) + P_{\mathcal{D}(\mathcal{X}_N)}(x \notin \mathcal{X}_{N_0})V_2(c_2).$$

To prove the theorem, it remains to show that $V_1(c_1) = V_1(c_2)$ and $V_2(c_1) = V_2(c_2)$.

By the definition of $\mathcal{F}/\left(c|_{\mathcal{X}_{N_0}}\right)$, we have

$$\begin{aligned}
V_1(c_1) &= \sum_{f \in \mathcal{F}/\left(c_1|_{\mathcal{X}_{N_0}}\right)} \mathbb{E}_{x \sim \mathcal{D}_1}\left[\ell\left(c_1(x), f(x)\right)\right] \\
&= \sum_{f \in \mathcal{F}/\left(c_1|_{\mathcal{X}_{N_0}}\right)} \mathbb{E}_{x \sim \mathcal{D}_1}\left[\ell\left(c_1(x), c_1(x)\right)\right] \\
&\overset{(a)}{=} \sum_{f \in \mathcal{F}/\left(c_2|_{\mathcal{X}_{N_0}}\right)} \mathbb{E}_{x \sim \mathcal{D}_1}\left[\ell\left(c_2(x), c_2(x)\right)\right] \\
&= \sum_{f \in \mathcal{F}/\left(c_2|_{\mathcal{X}_{N_0}}\right)} \mathbb{E}_{x \sim \mathcal{D}_1}\left[\ell\left(c_2(x), f(x)\right)\right] = V_1(c_2),
\end{aligned}$$

where the equality (a) is due to that $c_1(x) = c_2(x)$ for all $x \in \mathcal{X}_{N_0}$.

By the no-free-lunch theorem (Wolpert & Macready, 1997; Wolpert, 2002) for $\mathcal{X}_N \setminus \mathcal{X}_{N_0}$, $\mathcal{Y}$, and $\mathcal{F}_{\mathcal{X}_N \setminus \mathcal{X}_{N_0}, \mathcal{Y}}$, we have

$$\sum_{f \in \mathcal{F}_{\mathcal{X}_N \setminus \mathcal{X}_{N_0}, \mathcal{Y}}} \mathbb{E}_{x \sim \mathcal{D}_2}\left[\ell\left(c_1(x), f(x)\right)\right] = \sum_{f \in \mathcal{F}_{\mathcal{X}_N \setminus \mathcal{X}_{N_0}, \mathcal{Y}}} \mathbb{E}_{x \sim \mathcal{D}_2}\left[\ell\left(c_2(x), f(x)\right)\right].$$

Then we have

$$\begin{aligned}
V_2(c_1) &= \sum_{f \in \mathcal{F}/\left(c_1|_{\mathcal{X}_{N_0}}\right)} \mathbb{E}_{x \sim \mathcal{D}_2}\left[\ell\left(c_1(x), f(x)\right)\right] \\
&= \sum_{f \in \mathcal{F}_{\mathcal{X}_N \setminus \mathcal{X}_{N_0}, \mathcal{Y}}} \mathbb{E}_{x \sim \mathcal{D}_2}\left[\ell\left(c_1(x), f(x)\right)\right] \\
&= \sum_{f \in \mathcal{F}_{\mathcal{X}_N \setminus \mathcal{X}_{N_0}, \mathcal{Y}}} \mathbb{E}_{x \sim \mathcal{D}_2}\left[\ell\left(c_2(x), f(x)\right)\right] \\
&= \sum_{f \in \mathcal{F}/\left(c_2|_{\mathcal{X}_{N_0}}\right)} \mathbb{E}_{x \sim \mathcal{D}_2}\left[\ell\left(c_2(x), f(x)\right)\right] = V_2(c_2).
\end{aligned}$$

### C.2. Proof for Theorem 2

Let $z = z(x) = \mathrm{Proj}(x, V)$ and $\mathcal{Z}_{N_0} = \{\mathrm{Proj}(x, V) \mid x \in \mathcal{X}_{N_0}\}$. Let $\mathcal{G}'_{N_0, c}$ be the set of all RFM interpolators on $\mathcal{Z}_{N_0}$ for the concept $c(z)$, i.e.,

$$\mathcal{G}'_{N0, c} = \{f_{\mathrm{RFM}}(z; a) \mid f_{\mathrm{RFM}}(z; a) = c(z) \text{ for all } z \in \mathcal{Z}_{N_0}\},$$

where

$$f_{\mathrm{RFM}}(z; a) = \frac{1}{\sqrt{K}} \sum_{k=1}^{K} a_k \sigma(\langle w_k, z \rangle + b_k).$$

According to the proof of Theorem 3.8 in Abbe et al. (2023), the RFM model $f_{\mathrm{RFM}}(z; a)$ converges to the min-degree interpolator (w.r.t. the variable $z$) in $\mathcal{G}'_{N_0, c}$, i.e.,

$$f_{\mathrm{RFM}}(z; a_t) \to \underset{g' \in \mathcal{G}'_{N_0, c}}{\arg\min} \mathrm{DegP}_{B'}(g'), \quad \text{as } K \to \infty, t \to \infty,$$

where $B' = \{\chi_T(z)\}_{T \subseteq [r]}$.

By the definition of $z = z(x) = \text{Proj}(x, V)$, we have

$$f_{\text{RFMP}}^{V,K}(x; a_t) = f_{\text{RFM}}(z; a_t) \to \underset{g' \in \mathcal{G}'_{N_0,c}}{\arg\min} \, \text{DegP}_{B'}(g') = \arg \underset{g \in \mathcal{G}_{N_0,c,V}}{\min} \, \text{DegP}_{\mathcal{B}(V)}(g),$$

as $K \to \infty, t \to \infty$.

### C.3. Proof for Corollary 1

**Lemma 1.** *For any $f \in \mathcal{F}$, the min-degree interpolator $f^*$ on $\mathcal{X}_{N_0}$ satisfies $I(f^*) \subset [N_0]$.*

*Proof for Lemma 1.* Assume that there exist some $f \in \mathcal{F}$ such that the min-degreee interpolator $\hat{f}$ of $f$ on $\mathcal{X}_{N_0}$ does not satisfy $I(\hat{f}) \subset [N_0]$.

Let $\tilde{f}(x)$ be the function constructed by fixing all $x_i, i \notin [N_0]$ to 1 in $\hat{f}(x)$. By the construction, we have $\tilde{f}(x) = \hat{f}(x)$ for all $x \in \mathcal{X}_{N_0}$ and thus $\tilde{f}(x)$ is also an interpolator of $f$ on $\mathcal{X}_{N_0}$.

Since $I(\hat{f}) \not\subset [N_0]$, we have $\text{DegP}(\tilde{f}) < \text{DegP}(\hat{f})$. This contradicts the assumption that $\hat{f}$ is the min-degree interpolator. $\qquad\square$

By Lemma 1, for any $f \in \mathcal{F}$ such that $I(f) \not\subset [N_0]$, the min-degreee interpolator $\hat{f}$ is not identical to $f$ on $\mathcal{X}_N$ and thus does not achieves LDHD generalization from $\mathcal{X}_{N_0}$ to $\mathcal{X}_N$.

### C.4. Proof for Theorem 3

Let $U(\mathcal{X}_{N_0})$ be the uniform distribution over $\mathcal{X}_{N_0}$. The loss function is

$$L(P) = \mathbb{E}_{x \sim U(\mathcal{X}_{N_0})} \left[ \frac{1}{2} \left( f_{\text{PLAA}}^{\text{APE}}(x; P) - f_{\text{PLAA}}(x; A^*) \right)^2 \right]$$

$$= \mathbb{E}_{x \sim U(\mathcal{X}_{N_0})} \left[ \frac{1}{2} \left( x^\intercal (M_N^u \circ P^\intercal P) e_{n(x)} - x^\intercal A^* e_{n(x)} \right)^2 \right]$$

$$= \mathbb{E}_{x \sim U(\mathcal{X}_{N_0})} \left[ \frac{1}{2} \left( \left\langle x e_{n(x)}^\intercal, M_N^u \circ P^\intercal P \right\rangle - \left\langle x e_{n(x)}^\intercal, A^* \right\rangle \right)^2 \right]$$

$$\overset{(a)}{=} \mathbb{E}_{x \sim U(\mathcal{X}_{N_0})} \left[ \frac{1}{2} \left( \left\langle x e_{n(x)}^\intercal, M_N^u \circ P^\intercal P \right\rangle - \left\langle x e_{n(x)}^\intercal, M_N^u A^* \right\rangle \right)^2 \right]$$

$$= \mathbb{E}_{x \sim U(\mathcal{X}_{N_0})} \left[ \frac{1}{2} \left( \left\langle M_N^u \circ x e_{n(x)}^\intercal, P^\intercal P - A^* \right\rangle \right)^2 \right],$$

where (a) follows from the fact that $A^*$ is upper triangular.

**Lemma 2.** *For any matrix $Z \in \mathbb{R}^{N \times N}$, it holds that*

$$\mathbb{E}_{x \sim U(\mathcal{X}_{N_0})} \left[ \frac{1}{2} \left( \left\langle M_N^u \circ x e_{n(x)}^\intercal, Z \right\rangle \right)^2 \right] = \frac{1}{2} \| Q_{N_0} \circ Z \|_F^2,$$

*where*

$$Q_{N_0} = \begin{bmatrix} 2^{-\frac{N_0}{2}} & \cdots & \cdots & 2^{-\frac{1}{2}} & 0 & \cdots & 0 \\ 0 & 2^{-\frac{N_0-1}{2}} & \cdots & 2^{-\frac{1}{2}} & 0 & \cdots & 0 \\ \vdots & \ddots & \ddots & 2^{-\frac{1}{2}} & 0 & \cdots & 0 \\ 0 & \cdots & 0 & 2^{-\frac{1}{2}} & 0 & \cdots & 0 \\ 0 & \cdots & \cdots & 0 & 0 & \cdots & 0 \\ \vdots & \vdots & \vdots & \vdots & 0 & \cdots & 0 \\ 0 & \cdots & 0 & 0 & 0 & \cdots & 0 \end{bmatrix} \in \mathbb{R}^{N \times N},$$

*that is,*

$$(Q_{N_0})_{ij} = \begin{cases} 2^{\frac{N_0-j+1}{2}}, & 1 \le i \le j \le N_0, \\ 0, & \text{otherwise.} \end{cases}$$

*Proof for Lemma 2.* Note that

$$\mathbb{E}_{x \sim U(\mathcal{X}_{N_0})} \left[ \frac{1}{2} \left( \left\langle M_N^u \circ x e_{n(x)}^\mathsf{T}, Z \right\rangle \right)^2 \right] = \frac{1}{2^{N_0}} \sum_{k=1}^{N_0} \sum_{x:n(x)=k} \frac{1}{2} \left( \langle M_N^u \circ x e_k^\mathsf{T}, Z \rangle \right)^2$$

$$= \frac{1}{2^{N_0}} \sum_{k=1}^{N_0} \sum_{x:n(x)=k} \frac{1}{2} \left( \sum_{i=1}^{k-1} x_i Z_{ik} - Z_{kk} \right)^2$$

$$= \frac{1}{2^{N_0}} \sum_{k=1}^{N_0} \sum_{x:n(x)=k} \frac{1}{2} \sum_{i=1}^{k} Z_{ik}^2$$

$$\stackrel{(a)}{=} \frac{1}{2^{N_0}} \sum_{k=1}^{N_0} \frac{2^{k-1}}{2} \sum_{i=1}^{k} Z_{ik}^2$$

$$= \frac{1}{2} \sum_{k=1}^{N_0} \sum_{i=1}^{k} \left( 2^{-\frac{N_0-k+1}{2}} Z_{ik} \right)^2 = \frac{1}{2} \left\| Q_{N_0} \circ Z \right\|_F^2,$$

where (a) follows from the fact that $|\{x \mid n(x) = k\}| = 2^{k-1}$. $\qquad\square$

According to Lemma 2, we have

$$L(P) = \frac{1}{2} \left\| Q_{N_0} \circ (P^\mathsf{T} P - A^*) \right\|_F^2. \tag{3}$$

For any $0 < \alpha \le \alpha_0$, we have $L\left(P_\infty(\alpha)\right) = 0$ and thus

$$(M_N^u \circ P_\infty(\alpha)^\mathsf{T} P_\infty(\alpha))_{[N_0],[N_0]} = A^*_{[N_0],[N_0]}. \tag{4}$$

Let $p_i \in \mathbb{R}^{d_P}$ denote the $i$-th column of $P$. According to (3), we have

$$\frac{\partial L}{\partial p_i} = 0,$$

for all $i = N_0 + 1, \dots, N$. Therefore, $(p_i)_t = (p_i)_0$ for all $t$ and $i = N_0 + 1, \dots, N$.

Define

$$\tilde{A}^* := \begin{cases} (A^*)_{ij}, & 1 \le i \le j \le N_0, \\ 0, & \text{otherwise.} \end{cases}$$

Let $p_i^\alpha$ be the $i$-th column of $P_\infty(\alpha)$. We have

$$\left\| M_N^u \circ P_\infty^\mathsf{T}(\alpha) P_\infty(\alpha) - \tilde{A}^* \right\|_F^2 = \sum_{1 \le i \le j \le N} \left( (p_i^\alpha)^\mathsf{T} p_j^\alpha - \tilde{A}_{ij}^* \right)^2$$

$$\stackrel{(a)}{=} \sum_{j=N_0+1}^{N} \sum_{i=1}^{j} \left( (p_i^\alpha)^\mathsf{T} p_j^\alpha \right)^2$$

$$\le \sum_{j=N_0+1}^{N} \sum_{i=1}^{j} \|p_i^\alpha\|_2^2 \|p_j^\alpha\|_2^2$$

$$= \sum_{j=N_0+1}^{N} \sum_{i=1}^{N_0} \|p_i^\alpha\|_2^2 \|p_j^\alpha\|_2^2 + \sum_{j=N_0+1}^{N} \sum_{i=N_0+1}^{j} \|p_i^\alpha\|_2^2 \|p_j^\alpha\|_2^2.$$

where (a) follows from (4) and the definition of $\tilde{A}^*$.

Note that $\|p_i^\alpha\|_2^2 = A_{ii}^* \leq \|A^*\|_\infty$ for all $1 \leq i \leq N_0$ and $\|p_i^\alpha\|_2^2 = \alpha$ for all $N_0 + 1 \leq i \leq N$. Then we have

$$0 \leq \left\| M_N^u \circ P_\infty^\mathsf{T}(\alpha) P_\infty(\alpha) - \tilde{A}^* \right\|_F^2 \leq N(N - N_0)\alpha \|A^*\|_\infty + \frac{(N - N_0 + 1)(N - N_0)}{2}\alpha^2.$$

As $\alpha \to 0$, we have

$$0 \leq \left\| M_N^u \circ \hat{P}^\mathsf{T}\hat{P} - \tilde{A}^* \right\|_F^2 \leq 0,$$

which means $M_N^u \circ \hat{P}^\mathsf{T}\hat{P} = \tilde{A}^*$.

It remains to show that $f_{\text{PLAA}}^{\text{APE}}(x; \hat{P})$ is the min-degree interpolator w.r.t. $\mathcal{B}_N^{\text{PLAA}}$ on $\mathcal{X}_{N_0}$. Notice that

$$b_{ij}^{\text{PLAA}}(x) = \left\langle xe_{n(x)}^\mathsf{T}, E_{ij} \right\rangle.$$

Then we have

$$f_{\text{PLAA}}^{\text{APE}}(x; \hat{P}) = \sum_{1 \leq i \leq j \leq N} \tilde{A}_{ij}^* b_{ij}^{\text{PLAA}}(x) = \sum_{1 \leq i \leq j \leq N_0} A_{ij}^* b_{ij}^{\text{PLAA}}(x).$$

For any $g = \sum_{1 \leq i \leq j \leq N} A_{ij} b_{ij}^{\text{PLAA}}(x) \in \mathcal{G}_{N_0, A^*}^{\text{PLAA}}$, we have $A_{ij} = A_{ij}^*$ for all $1 \leq i \leq j \leq N_0$. Since $A_{ij}^2 \geq \tilde{A}_{ij}^2$ for all $1 \leq i \leq j \leq N$, we have

$$\text{DegP}_{\mathcal{B}_N^{\text{PLAA}}} \left( f_{\text{PLAA}}^{\text{APE}}(x; \hat{P}) \right) \leq \text{DegP}_{\mathcal{B}_N^{\text{PLAA}}} (g).$$

Hence, $f_{\text{PLAA}}^{\text{APE}}(x; \hat{P})$ is the min-degree interpolator w.r.t. $\mathcal{B}_N^{\text{PLAA}}$ on $\mathcal{X}_{N_0}$.

### C.5. Proof for Theorem 4

Note that

$$f_{\text{PLAA}}^{\text{GRPE},\mathcal{U}}(x; p) = x^\mathsf{T} \left( \sum_{k=1}^r p_k U_k \right) e_{n(x)} = \left\langle xe_{n(x)}^\mathsf{T}, \sum_{k=1}^r p_k U_k \right\rangle.$$

Let $D_{N_0}$ be the size of the training set in $\mathcal{X}_{N_0}$. Then the loss can be represented as

$$L(p) = \frac{1}{2D_{N_0}} \sum_{x \in \mathcal{X}_{N_0}} \left( \left\langle xe_{n(x)}^\mathsf{T}, \sum_{k=1}^r p_k U_k \right\rangle - c^*(x) \right)^2.$$

Without loss of generality, we use the notation of gradient flow in this proof. Then we have

$$\dot{A} = -\frac{1}{D_{N_0}} \sum_{x \in \mathcal{X}_{N_0}} \left( \left\langle xe_{n(x)}^\mathsf{T}, A \right\rangle - c^*(x) \right) \sum_{k=1}^r \left\langle xe_{n(x)}^\mathsf{T}, U_k \right\rangle U_k.$$

Since $p(0) = 0$, we have $A(0) = 0$ and thus $A(t) \in \text{span} \left\{ \sum_{k=1}^r \left\langle xe_{n(x)}^\mathsf{T}, U_k \right\rangle U_k \right\}_{x \in \mathcal{X}_{N_0}}$. Then the convergence point $\hat{A}$ can be represented as $\hat{A} = \sum_{x \in \mathcal{X}_{N_0}} \hat{a}(x) \sum_{k=1}^r \left\langle xe_{n(x)}^\mathsf{T}, U_k \right\rangle U_k$. The convergence function $\hat{f}$ is

$$\hat{f}(x) = \sum_{x' \in \mathcal{X}_{N_0}} \hat{a}(x') \sum_{k=1}^r \left\langle x'e_{n(x')}^\mathsf{T}, U_k \right\rangle \left\langle xe_{n(x)}^\mathsf{T}, U_k \right\rangle$$

$$= \sum_{k=1}^r \sum_{x' \in \mathcal{X}_{N_0}} \hat{a}(x') \left\langle x'e_{n(x')}^\mathsf{T}, U_k \right\rangle \left\langle xe_{n(x)}^\mathsf{T}, U_k \right\rangle$$

**Lemma 3.** *For any upper triangle matrix $U \in \mathbb{R}^{N \times N}$, we have*

$$\langle x e_{n(x)}^\mathsf{T}, U \rangle = \sum_{1 \leq i \leq j \leq N} U_{ij} b_{ij}^{PLAA}(x).$$

*Proof for Lemma 3.* We first prove that $\langle x e_{n(x)}^\mathsf{T}, E_{mn} \rangle = b_{ij}^{\mathrm{PLAA}}(x)$ for any $1 \leq i \leq j \leq N$. Notice that

$$\langle x e_{n(x)}^\mathsf{T}, E_{ij} \rangle = \begin{cases} x_i & x_j = -1 \wedge x_{j+1} = 1 \wedge \cdots \wedge x_N = 1, \\ 0 & \text{otherwise.} \end{cases}$$

Therefore, we have

$$\begin{aligned} \langle x e_{n(x)}^\mathsf{T}, E_{ij} \rangle &= x_i \cdot \frac{1 - x_j}{2} \cdot \frac{1 + x_{j+1}}{2} \cdot \ldots \cdot \frac{1 + x_N}{2} \\ &= \begin{cases} -\frac{1 - x_j}{2} \prod_{k=j+1}^N \frac{1 + x_k}{2}, & i = j, \\ x_i \frac{1 - x_j}{2} \prod_{k=j+1}^N \frac{1 + x_k}{2}, & i < j. \end{cases} \\ &= b_{ij}^{\mathrm{PLAA}}(x). \end{aligned}$$

For any upper triangle matrix $U = \sum_{1 \leq i \leq j \leq N} U_{ij} E_{ij}$, we have

$$\langle x e_{n(x)}^\mathsf{T}, U \rangle = \langle x e_{n(x)}^\mathsf{T}, \sum_{1 \leq i \leq j \leq N} U_{ij} E_{ij} \rangle = \sum_{1 \leq i \leq j \leq N} U_{ij} \langle x e_{n(x)}^\mathsf{T}, E_{ij} \rangle = \sum_{1 \leq i \leq j \leq N} U_{ij} b_{ij}^{\mathrm{PLAA}}(x).$$

$\square$

Define $b_k(x) := \langle x e_{n(x)}^\mathsf{T}, U_k \rangle$. By Lemma 3, we have $b_k(x) = \sum_{1 \leq i \leq j \leq N} (U_k)_{ij} b_{ij}^{\mathrm{PLAA}}(x)$. Then the convergence function $\hat{f}$ can be represented as

$$\hat{f}(x) = \sum_{k=1}^r \sum_{x' \in \mathcal{X}_{N_0}} \hat{a}(x') b_k(x') b_k(x) = \sum_{k=1}^r \hat{p}_k b_k(x),$$

where $\hat{p}_k = \sum_{x' \in \mathcal{X}_{N_0}} \hat{a}(x') b_k(x')$.

For any interpolator $g(x) = \sum_{k=1}^r p_k b_k(x)$ on $\mathcal{X}_{N_0}$:

- If $b_k(x) = 0$ for all $x \in \mathcal{X}_{N_0}$, we have $\hat{p}_k = 0$ and $p_k^2 \geq 0 = \hat{p}_k^2$;

- If $b_k(x) \neq 0$ for some $x \in \mathcal{X}_{N_0}$, we have $\hat{p}_k = p_k$ and thus $\hat{p}_k^2 = p_k^2$. To show this, without loss of generality, we suppose that for $1 \leq k \leq m$, there exists some $x \in \mathcal{X}_{N_0}$ such that $b_k(x) \neq 0$, and for $m + 1 \leq k \leq r$, $b_k(x) = 0$ for all $x \in \mathcal{X}_{N_0}$. It suffices to show that the following equation has a unique solution:

$$\begin{bmatrix} b_1(x_1) & \ldots & b_m(x_1) \\ \vdots & \vdots & \vdots \\ b_1(x_M) & \ldots & b_m(x_M) \end{bmatrix} \begin{bmatrix} p_1 \\ \vdots \\ p_m \end{bmatrix} = \begin{bmatrix} c^*(x_1) \\ \vdots \\ c^*(x_M) \end{bmatrix},$$

where $M = |\mathcal{X}_{N_0}|$ and $\mathcal{X}_{N_0} = \{x_1, \ldots, x_M\}$. Since $p_1 = \hat{p}_1, \ldots, p_m = \hat{p}_m$ is a solution, it remains to prove the uniqueness. Let

$$B_m = \begin{bmatrix} b_1(x_1) & \ldots & b_m(x_1) \\ \vdots & \vdots & \vdots \\ b_1(x_M) & \ldots & b_m(x_M) \end{bmatrix}.$$

It suffices to show $\mathrm{rank}(B_m) = m$. Note that $(B_m^\mathsf{T} B_m)_{ii} = \sum_{x \in \mathcal{X}_{N_0}} b_i(x)^2 > 0$ and $(B_m^\mathsf{T} B_m)_{ij} = \sum_{x \in \mathcal{X}_{N_0}} b_i(x) b_j(x) = 0$. We have $\mathrm{rank}(B_m^\mathsf{T} B_m) = m$ and thus $\mathrm{rank}(B_m) = m$.

Hence, for any interpolator $g$ on $\mathcal{X}_{N_0}$, we have

$$\mathrm{DegP}_{\mathcal{B}_{\mathrm{PLAA}}^{\mathrm{GRPE},\mathcal{U}}}(\hat{f}) \leq \mathrm{DegP}_{\mathcal{B}_{\mathrm{PLAA}}^{\mathrm{GRPE},\mathcal{U}}}(g),$$

or equivalently,

$$f_{\mathrm{PLAA}}^{\mathrm{GRPE},\mathcal{U}}(x; p_t) \to \arg \min_{g \in \mathcal{G}_{N_0, p^*}^{\mathrm{GRPE},\mathcal{U}}} \mathrm{DegP}_{\mathcal{B}_{\mathrm{PLAA}}^{\mathrm{GRPE},\mathcal{U}}}(g), \quad \text{as } t \to \infty.$$

## C.6. Proof for Corollary 2

By Theorem 4, the PLAA with GRPE achieves LHDH generalization from $\mathcal{X}_{N_0}$ to $\mathcal{X}_N$ if and only if the target concept $c(x)$ is the min-degree interpolator w.r.t. the linearly independent set $\mathcal{B}_{\mathrm{PLAA}}^{\mathrm{GRPE},\mathcal{U}}$. Therefore, it is equivalent to prove the target concept $c(x)$ is the min-degree interpolator w.r.t. the linearly independent set $\mathcal{B}_{\mathrm{PLAA}}^{\mathrm{GRPE},\mathcal{U}}$ if and only if

$$\left\{ k \mid (U_k)_{[N_0],[N_0]} = 0 \right\} \subseteq \left\{ k \mid c_k = 0 \right\}.$$

We denote $\left\{ k \mid (U_k)_{[N_0],[N_0]} = 0 \right\}$ by $\mathcal{K}_1$ and $\{ k \mid c_k = 0 \}$ by $\mathcal{K}_2$ in this proof.

We first show the sufficiency. Assume that $c(x)$ is not the min-degree interpolator w.r.t. the linearly independent set $\mathcal{B}_{\mathrm{PLAA}}^{\mathrm{GRPE},\mathcal{U}}$ on $\mathcal{X}_{N_0}$. In other words, there exists an interpolator $\tilde{c}(x) = \sum_{k=1}^{r} \tilde{c}_k \sum_{1 \leq i \leq j \leq N} (U_k)_{ij} b_{ij}^{\mathrm{PLAA}}(x)$ on $\mathcal{X}_{N_0}$ such that

$$\mathrm{DegP}_{\mathcal{B}_{\mathrm{PLAA}}^{\mathrm{GRPE},\mathcal{U}}}(\tilde{c}) < \mathrm{DegP}_{\mathcal{B}_{\mathrm{PLAA}}^{\mathrm{GRPE},\mathcal{U}}}(c).$$

Since both $c(x)$ and $\tilde{c}(x)$ are interpolators on $\mathcal{X}_{N_0}$, we have

$$\sum_{k=1}^{r} c_k (U_k)_{[N_0],[N_0]} = \sum_{k=1}^{r} \tilde{c}_k (U_k)_{[N_0],[N_0]},$$

or equivalently,

$$\sum_{k \notin \mathcal{K}_1} c_k (U_k)_{[N_0],[N_0]} = \sum_{k \notin \mathcal{K}_1} \tilde{c}_k (U_k)_{[N_0],[N_0]}.$$

Since $(U_i)_{kl}(U_j)_{kl} = 0$ for all $i \neq j$ and $1 \leq k, l \leq N$, we have $\left\{ (U_k)_{[N_0],[N_0]} \right\}_{k \notin \mathcal{K}}$ are linearly independent. This implies $c_k = \tilde{c}_k$ for all $k \notin \mathcal{K}_1$.

Since $\mathcal{K}_1 \subseteq \mathcal{K}_2$, we have $c_k = 0$ and thus $c_k^2 \leq \tilde{c}_k^2$ for all $k \in \mathcal{K}_1$. Hence, we have

$$\mathrm{DegP}_{\mathcal{B}_{\mathrm{PLAA}}^{\mathrm{GRPE},\mathcal{U}}}(c) \leq \mathrm{DegP}_{\mathcal{B}_{\mathrm{PLAA}}^{\mathrm{GRPE},\mathcal{U}}}(\tilde{c}),$$

which contradicts the assumption.

We then prove the necessity. Assume that $\mathcal{K}_1 \nsubseteq \mathcal{K}_2$. Then there exists some $k_0 \in \mathcal{K}_1$ but $k_0 \notin \mathcal{K}_2$. Define $\tilde{\tilde{c}}(x) = \sum_{k=1}^{r} \tilde{\tilde{c}}_k \sum_{1 \leq i \leq j \leq N} (U_k)_{ij} b_{ij}^{\mathrm{PLAA}}(x)$ where

$$\tilde{\tilde{c}}_k = \begin{cases} 0, & k = k_0, \\ c_k, & k \neq k_0. \end{cases}$$

By the definition of $\mathcal{K}_1$, $\tilde{\tilde{c}}(x) = c(x)$ for all $x \in \mathcal{X}_{N_0}$ and thus $\tilde{\tilde{c}}(x)$ is also an interpolator on $\mathcal{X}_0$. Note that $c_{k_0} \neq 0$. By the definition of $\tilde{\tilde{c}}(x)$, we have

$$\mathrm{DegP}_{\mathcal{B}_{\mathrm{PLAA}}^{\mathrm{GRPE},\mathcal{U}}}(\tilde{\tilde{c}}) \leq \mathrm{DegP}_{\mathcal{B}_{\mathrm{PLAA}}^{\mathrm{GRPE},\mathcal{U}}}(c),$$

which contradicts that the target concept $c(x)$ is the min-degree interpolator w.r.t. the linearly independent set $\mathcal{B}_{\mathrm{PLAA}}^{\mathrm{GRPE},\mathcal{U}}$ on $\mathcal{X}_{N_0}$.

# D. Experiments

## D.1. Detailed Explanation of RPE-Square

The design of RPE-Square follows the principle: when devising position embeddings for length generalization, one needs to consider both the LDHD generalization in the latent space and the nuisance of the data format in the input sequence space. This principle is derived from the LDHD generalization perspective for length generalization, illustrating the insights of LDHD generalization for practical models.

To further elaborate, we show how RPE-Square is constructed in the guidance of the proposed principle to achieve length generalization of the URF addition (with CoT).

- **LDHD generalization in the latent space**. For the addition, the LDHD generalization in the latent space can be handled by RPE.

- **The data format nuisance of URF**. To handle the data format, we need to consider the mapping between a latent variable and its URF string, i.e., how the latent variable $h_n = [(x_0, y_0, z_0), \ldots, (x_{t-1}, y_{t-1}, z_{t-1}), (x_t, y_t, *), \ldots, (x_{n-1}, y_{n-1}, *)]$ can be recovered from the corresponding URF string "$[\texttt{BOS}]\texttt{x}_0 \ldots \texttt{x}_{\texttt{n}_\texttt{x}-1} + \texttt{y}_0 \ldots \texttt{y}_{\texttt{n}_\texttt{y}-1} = \texttt{z}_0 \ldots \texttt{z}_{t-1}$" (to predict $\texttt{z}_\texttt{t}$). The URF mapping can be described as follows: concatenate "$[\texttt{BOS}]$", the first elements of the dimensions (up to the "highest" nonzero element), a "+", the second elements of the dimensions (up to the "highest" nonzero element), a "=", and the third elements of the dimensions (up to the "highest" non-"*" element).

  According to the URF mapping, we notice that the "position" of an element in the latent variable can be identified by its relative distances to "$[\texttt{BOS}]$", "+", and "=". (Here, the relative distance from the position $i$ to the position $j$ is $i - j$.). Denote the tuple of the relative distances from some token $s$ to "$[\texttt{BOS}]$", "+", "=" by $(n_1(s), n_2(s), n_3(s))$. Then we have

$$s = \begin{cases} x_{n_1(s)-1}, & n_1(s) > 0, n_2(s) < 0, n_3(s) < 0, \\ y_{n_2(s)-1}, & n_1(s) > 0, n_2(s) > 0, n_3(s) < 0, \\ z_{n_3(s)-1}, & n_1(s) > 0, n_2(s) > 0, n_3(s) > 0. \end{cases}$$

  Note that $z_k$ can be determined by $x_{k-1}$, $x_k$, $y_{k-1}$, $y_k$, and $z_k$ (we ignore the boundary cases in this discussion for simplicity). To predict the next token of some $s$ where $n_1(s) > 0, n_2(s) > 0, n_3(s) > 0$, we need to identify the elements $s_1, s_2, s_3, s_4, s_5$ such that

$$\begin{cases} n_1(s_1) = n_3(s) - 1, & n_2(s_1) < 0, & n_3(s_1) < 0, \\ n_1(s_2) = n_3(s), & n_2(s_2) < 0, & n_3(s_2) < 0, \\ n_2(s_3) = n_3(s) - 1, & n_1(s_3) > 0, & n_3(s_3) < 0, \\ n_2(s_4) = n_3(s), & n_1(s_4) > 0, & n_3(s_4) < 0, \\ n_3(s_5) = n_3(s) - 1, & n_1(s_5) > 0, & n_2(s_5) > 0. \end{cases}$$

  Therefore, to address the URF data format nuisance, the position embedding can consider the relative distances to some tokens.

## D.2. Experiment Details

### D.2.1. UNALIGNED COPY

An instance of the unaligned copy task is like

$$\texttt{b}\,\texttt{x}_0 \ldots \texttt{x}_{\texttt{n}-1} = \texttt{x}_0 \ldots \texttt{x}_{\texttt{n}-1}\,\texttt{e}.$$

The model is given the input $\texttt{bx}_0 \ldots \texttt{x}_{\texttt{n}-1} =$ and expected to output the copy of $\texttt{x}_0 \ldots \texttt{x}_{\texttt{n}-1}$. We use "b" and "e" instead of "$[\texttt{BOS}]$" and "$[\texttt{EOS}]$" for a simpler implementation with the GPT-2 tokenizer.

We sample 2000 $n$-length instances for each $n = 1, \ldots, 5$ as the training data. In the evaluation, we examine the learned models on instances of length $1 - 10$. We train GPT-2 with key-only RPE and RPE-Square, respectively. The model is trained by AdamW with the cosine scheduler, where the initial learning rate is 0.0005, the weight decay is 1.0, the warmup ratio is 0.05, the gradient accumulation step is 2, and the per-device training batch size is 256. We set the training steps to

10000 but early stop at step 1000 as the model with RPE-Square has achieved nearly perfect length generalization while the model with RPE shows almost no length generalization then.

From the perspective of LDHD generalization, the latent variable corresponding to the input $\mathtt{b}x_0 \ldots x_{n-1} = x_0 \ldots x_k$ is

$$[(x_0, x_0), \ldots, (x_k, x_k), (x_{k+1}, *), \ldots, (x_N, x_N)].$$

The LDHD generalization in the latent space can be effectively addressed by RPE. The data format mapping can be handled by considering the relative distance to the tokens "b" and "=". Therefore, RPE-Square is expected to work for the length generalization of the unaligned copy. RPE does not properly deal with the unaligned data format and thus could fail to achieve length generalization in this scenario.

### D.2.2. URF ADDITION

For the URF $n$-addition training data, we first sample the lengths of two addends uniformly from $\{1, \ldots, n\} \times \{1, \ldots, n\}$. For two addends of lengths $(n_1, n_2)$, we then samples from $[10^{n_1-1}, 10^{n_1} - 1] \times [10^{n_2-1}, 10^{n_2} - 1]$ (If $n_i = 0$, then the corresponding sample interval is $[0, 9]$). This is to guarantee the addends are length-uniform. For the addition $x_{n_1-1} \ldots x_0 + y_{n_2-1} \ldots y_0 = z_{n_3} z_{n_3-1} \ldots z_0$, the training instance is

$$\mathtt{b}\, x_0\, \ldots\, x_{n_1-1}\, +\, y_0\, \ldots\, y_{n_2-1}\, =\, z_0\, \ldots\, z_{n_3-1}\, z_{n_3}\, \mathtt{e}.$$

Here, we add spaces between the characters to ensure each is tokenized separately. In our experiments, we train with 10000 URF 4-addition samples.

We choose GPT-2 with key-only position embeddings as our model. For the RPE and RPE-Square settings, we augment the GPT-2 model with RPE and RPE-Square, respectively. The implementation is adapted from HuggingFace (Wolf et al., 2020).

We train the models by AdamW, with the initial learning rate 0.0005, the weight decay 1.0, and the cosine scheduler. The warmup ratio is 0.05. The gradient accumulation step is 2. The per-device training batch size is 128. We train the models for 20000 steps and 200000 steps. The experiments are run on a server with Ubuntu. The models are trained on two NVIDIA GeForce RTX 3090 GPUs.

## E. Additional Experiments

### E.1. Additional Evaluations of RPE-Square

We present more evaluation results of RPE-Square by comparing it against more different position embeddings in additional tasks. Concretely, we consider three extra tasks: ParityCoT, Multiplication (1 * N), and Division (1 * N). Across all tasks (including Addition and Copy), we compare RPE-Square with five position embeddings: RPE (Shaw et al., 2018), RoPE (Su et al., 2024), NoPE (Kazemnejad et al., 2024), ALiBi (Press et al., 2021), and Abacus (McLeish et al., 2024). We keep the hyperparameter setting for the addition position embeddings in Addition and Copy: the model is trained by AdamW with the cosine scheduler, the initial learning rate is 0.0005, the weight decay is 1.0, the warmup ratio is 0.05, the gradient accumulation step is 2, and the per-device training batch size is 256. In the three extra tasks, we choose the initial learning rate as 0.0005 and keep the other hyperparameters the same.

Figures 3-7 presents the experimental results. Both RPE-Square and Abacus achieve non-trivial length generalization performance in the five tasks, and the overall evaluation results of RPE-Square are better than those of Abacus. The other four position embeddings do not achieve good length generalization performance. The reason, from our LDHD perspective, is that RPE-Square and Abacus succeed in handling the unaligned data format, while the others do not.

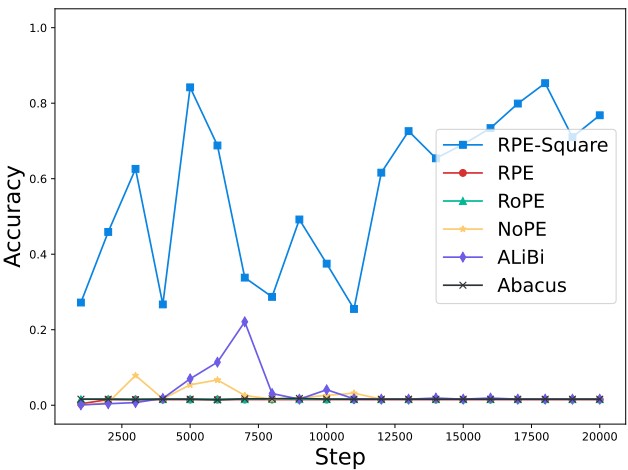

*Figure 3.* Addition. The models are trained on URF 4-addition and tested on URF 5-addition

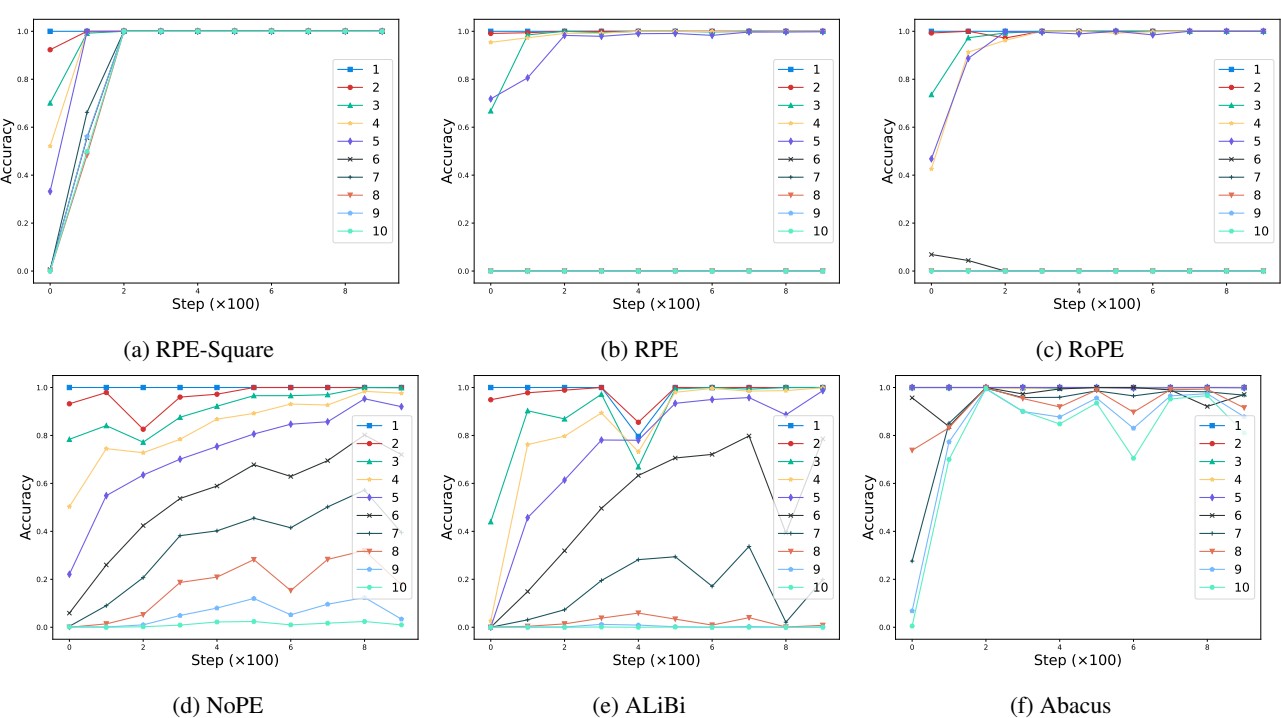

*Figure 4.* Copy. The models are trained on scales 1-5 and tested on scales 1-10.

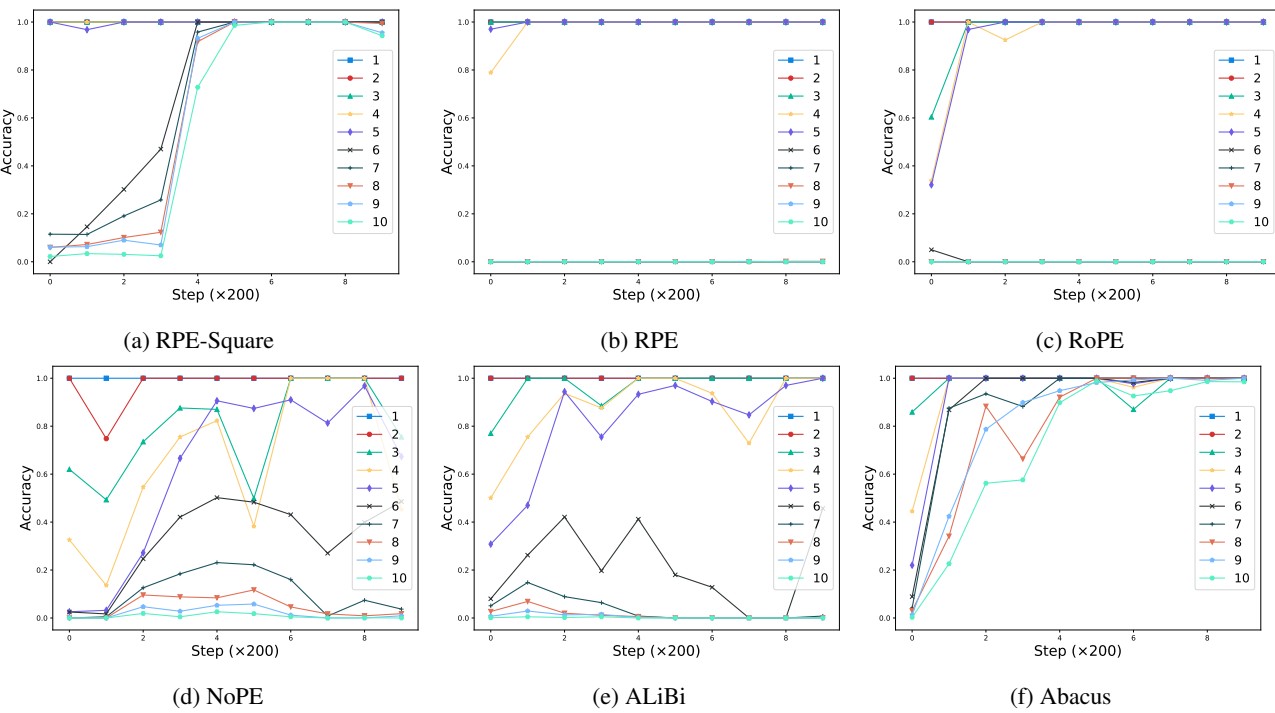

*Figure 5.* ParityCoT. The models are trained on scales 1-5 and tested on scales 1-10. An instance is like "$[\texttt{BOS}]\texttt{x}_0 \ldots \texttt{x}_{\texttt{n}-1} = \texttt{y}_0 \ldots \texttt{y}_{\texttt{n}-1}[\texttt{EOS}]$", where $y_0 = x_0$ and $y_k = x_k \oplus y_{k-1}$ for $k = 1, \ldots, n-1$.

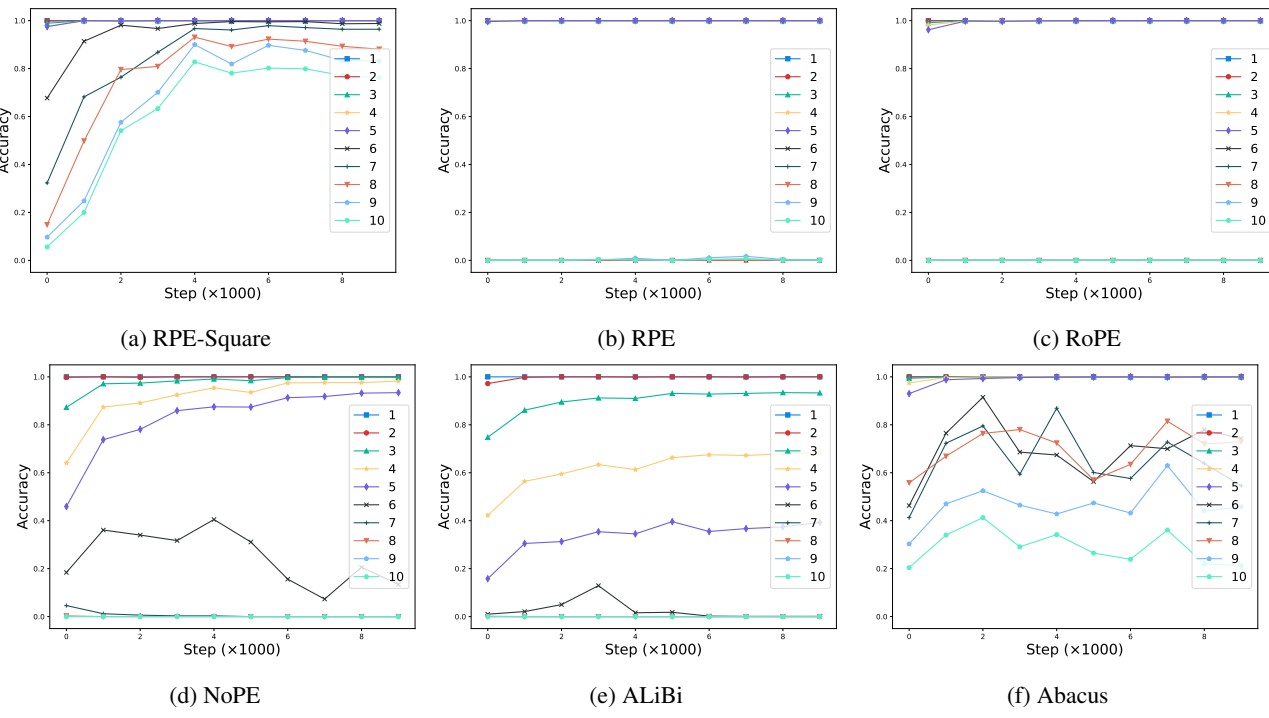

*Figure 6.* Multiplication $(1 * N)$. The models are trained on scales 1-5 and tested on scales 1-10. An instance is like "$[\texttt{BOS}]\texttt{x} * \texttt{y}_0 \ldots \texttt{y}_{\texttt{n}-1} = \texttt{z}_0 \ldots \texttt{z}_{\texttt{n}}[\texttt{EOS}]$".

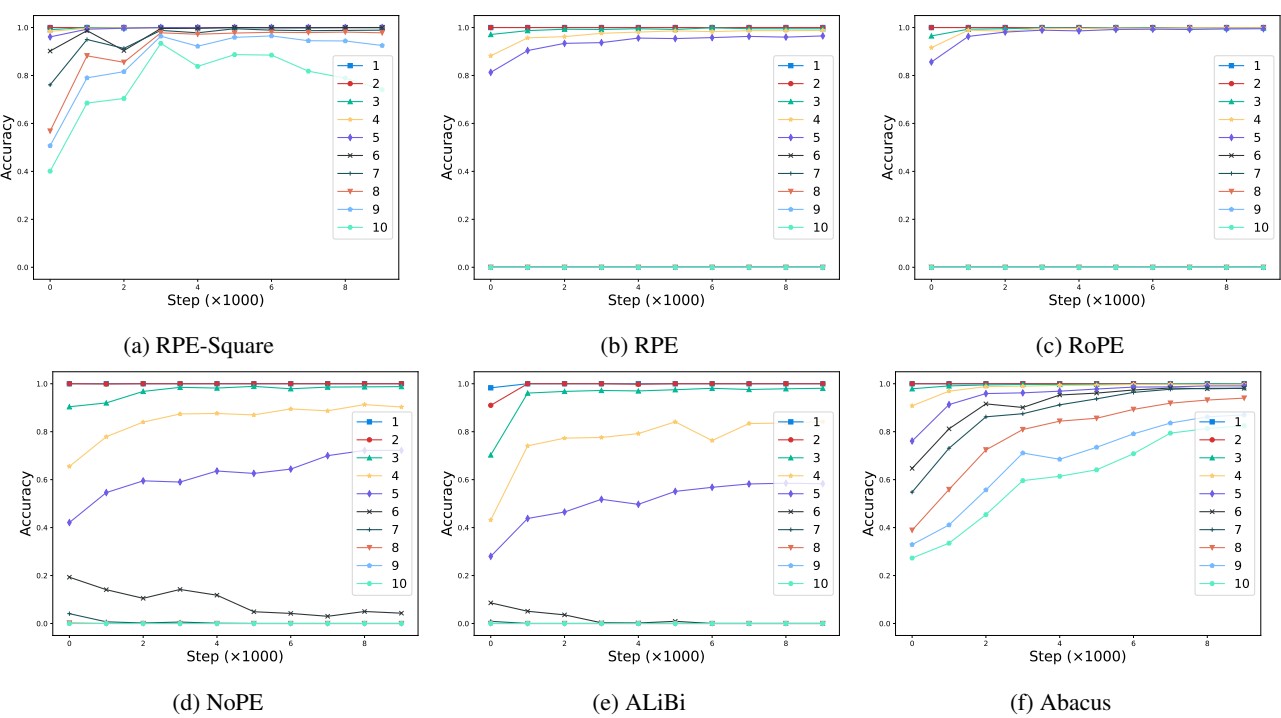

(a) RPE-Square         (b) RPE         (c) RoPE

(d) NoPE         (e) ALiBi         (f) Abacus

*Figure 7.* Division $(N/1)$. The models are trained on scales 1-5 and tested on scales 1-10. An instance is like "$[\texttt{BOS}]\texttt{x} \setminus \texttt{y}_{n-1} \ldots \texttt{y}_0 = \texttt{z}_{n-1} \ldots \texttt{z}_0[\texttt{EOS}]$", where $z = y//x$.

### E.2. Additional Application of the Position Embedding Design Principle

In this subsection, we present another application of the position embedding design principle besides RPE-Square. We consider a task called AdditionMod10, which computes the sum of the addends modulo 10, in the unaligned format. An instance takes the form

$$[\texttt{BOS}]\texttt{x}_0 \ldots \texttt{x}_{n-1} + \texttt{y}_0 \ldots \texttt{y}_{n-1} = \texttt{z}[\texttt{EOS}],$$

where $z = (x + y) \bmod 10$. Due to the modulo operation, the output depends solely on the first digits of the addends. This means the positional embedding need to capture the absolute relative distances to specific tokens (namely, "[BOS]" and "+"). The "absolute value" handles the LDHD generalization in the latent space, and "the relative distances to some special tokens" deals with the unaligned data format. Following the same design principle used in RPE-Square, we introduce a new position embedding called RPE-Absolute to encode "the absolute value of the relative distances to some special tokens". The expression of RPE-Absolute$_{i,j}$ is

$$\sum_{1 \leq k \leq i} \frac{\exp\left((W_Q x_i)^\intercal (W_K x_k)\right)}{\sum_{1 \leq k' \leq i} \exp\left((W_Q x_i)^\intercal (W_K x_{k'})\right)} R_{i-k},$$

where $W_K, W_Q, R$ are the learnable parameters.

The results are shown in Figure 8. We compare GPT-2 with RPE-Absolute against GPT-2 with RPE. The training hyperparameters are identical to those in the experiments of RPE-Square with the initial learning rate $0.0001$. The results demonstrate that RPE-Absolute achieves length generalization, while RPE only generalizes within the training lengths. This finding further supports the effectiveness of our position embedding design principle.

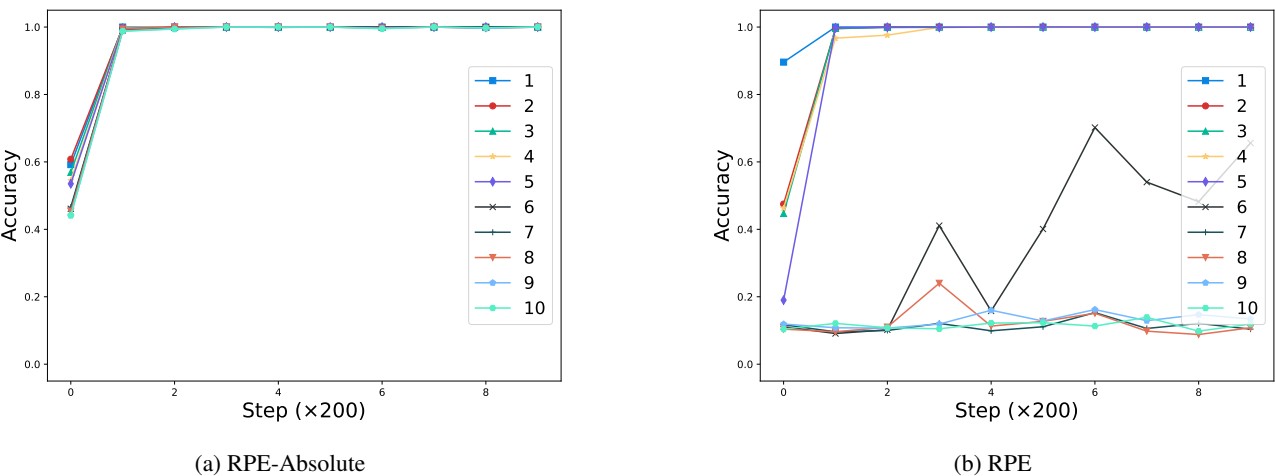

(a) RPE-Absolute

(b) RPE

*Figure 8.* AdditionMod10. The models are trained on scales 1-5 and tested on scales 1-10.

