# OpenReview forum: "Low-Dimension-to-High-Dimension Generalization and Its Implications for Length Generalization"
_ICML.cc/2025/Conference — ICML 2025 poster_

### Official Review · Reviewer_9ohk · 2025-02-26

**Overall Recommendation:** 3

**Summary:**

This paper conducts a theoretical analysis of the problem of low-dimension-to-high-dimension generalization, with an application to the problem of length generalization. The main theoretical results are based on Boolean function analysis and extend the analysis in Abbe et al. (2023) by considering different functional bases in the Boolean function space and analyzing more neural network models, including random feature models with projection and Position-Only Linear Attention with Advice (PLAA) models with Absolute Position Embedding (APE) and Generalized Absolute Position Embedding (GAPE). The paper shows that gradient descent on these models results in min-degree solutions with regard to different functional bases. On the algorithm side, the paper proposes RPE-Square, an ad-hoc improvement of Relative Position Embedding (RPE), and demonstrates its effectiveness on length generalization problems including unaligned copy and URF addition.

**Post-rebuttal update:** The authors have addressed some of my concerns. At this point, my main criticisms of the paper are (1) the limited implications of the formulation and theoretical analysis (compared with the prior work by Abbe et al. (2023)), and (2) the overall presentation (see my rebuttal comment for details). The authors have responded to both points in the rebuttal, but not to a very satisfactory level. I decided to keep my original score but would let AC and other reviewers decide whether this paper should be considered to reach the acceptance bar of ICML.

**Claims And Evidence:**

Most of the claims made in the paper are supported by theoretical or empirical evidence. However, I did not understand the claims on Chain-of-Thought (CoT) prompting in Sec. 5.1. See "Questions for authors" for more details.

**Essential References Not Discussed:**

To my knowledge, most of the related works are properly cited and discussed.

**Experimental Designs Or Analyses:**

Experimental designs are OK, though it would be better if the authors could also compare RPE-square with more position embedding methods (e.g., RoPE) in the experiments.

**Methods And Evaluation Criteria:**

The proposed RPE-Square is evaluated on two representative length generalization tasks including unaligned copy and URF addition. I am not an expert in benchmarking length generalization, but given the scope and scale of the considered tasks, they should mainly serve as proof-of-concept demonstrations at this stage. Since the main contents of the paper are of theoretical nature, I think the current evaluation is acceptable.

**Other Comments Or Suggestions:**

In general, at this point my overall evaluations of this paper are as follows:

- The theoretical contributions of the paper are rather incremental;
- The proposed method RPE-Square is interesting but is only evaluated in limited settings;
- The theory and algorithm seem disconnected.

I am now between weak accept and weak reject and decided to choose weak accept. Yet, I may also lower my rating if my main concerns are not well addressed in the rebuttal.

**Other Strengths And Weaknesses:**

*Strengths:* Theoretical results are clearly presented.

*Weaknesses:*
- The paper is not very easy to read, especially for those who are not familiar with the prior work by Abbe et al. (2023).
- Many definitions and results heavily build on those in Abbe et al. (2023), e.g., degree profile, min-degree interpolators, Theorem 2, and Corollary 1. Overall I think the theoretical contributions of the paper are somewhat incremental.
- Many parts of the paper, in my opinion, are kind of disconnected:
  - I did not get how Abstraction 1 is used in the theoretical results/proofs.
  - The theory and the algorithm seem not connected; in fact, I think that Theorems 3 and 4 are not relevant to the proposed RPE-Square (which seems to have nothing to do with functional bases induced by the position embedding), nor are they relevant to Example 1 (regarding the mismatch between the problem scale and the input length) which serves as the main motivation of the paper.
- I think that the main theoretical contributions of the paper, i.e., introducing min-degree interpolators under different bases, can only address very limited length generalization problems: for example, the target function in Example 3 only has degree 1. From my angle, introducing additional projections/linear operations cannot solve length generalization problems with higher-degree target functions, e.g., the one considered in Theorem 5.1 of Abbe et al. (2023).

**Questions For Authors:**

- I did not understand Sec. 5.1: how did you determine the latent space of each instance? Why should the latent space take the form in the paper?

- It appears to me that the formulation of LDHD generalization (Def. 1) is quite similar to length generalization (the main difference is that LDHD considers different dimensions in train/test data subspaces, while length generalization replaces the notion of "dimension" to "length"). Are there any other applications of this formulation except length generalization?

**Relation To Broader Scientific Literature:**

Prior work by Abbe et al. (2023) has shown that several types of NNs converge to min-degree interpolators when trained by gradient descent; this work extends this result by showing that other types of NNs, including simplified attention models with position embeddings, may converge to min-degree interpolations under different bases in the function space.

The proposed RPE-Square is an ad-hoc improvement to RPE in mathematical reasoning problems.

**Theoretical Claims:**

I reviewed some proofs in the appendix and did not find major issues.

---

> ### Author Rebuttal · Authors · 2025-04-01
>
> Thank you for your effort and insightful comments. We respond to your main concerns below.
> All experiments results are in the anonymous link: https://www.dropbox.com/scl/fi/52t23nfzev1lo1sq5dmyj/ICML_2025_5857_Rebuttal.pdf?rlkey=5nde6aampze744klvn0rsb3mp&st=37lg3nzo&dl=0.
>
> 1. *The theoretical contributions of the paper are rather incremental.*
>
> Our main contribution is the LDHD generalization formulation (Abstraction 1), which offers a more precise characterization of length generalization by separating it into two challenges: (1) LDHD generalization in the latent space (which is inherent), and (2) data format nuisance. Under the formulation, we can derive the fundamental limitation of length generalization, explain the effectiveness of certain methods, and design novel position embeddings for challenging length generalization problem.
>
> The theoretical results in Section 4 demonstrates how different model designs can introduce different inductive bias to handle LDHD generalization, only related to the LDHD generalization aspect in Abstraction 1.
>
> While our Boolean analysis in Section 4 is technically inspired by the remarkable work of Abbe et al. (2023), our core novelty lies in Abstraction 1 and its implications, not in extending prior Boolean analysis.
>
> 2. *The proposed method RPE-Square is interesting but is only evaluated in limited settings.*
>
> We evaluate RPE-Square in three additional tasks: Parity (with CoT), Multiplication (1 * N), and Division (N / 1). RPE-Square achieves length generalization in all the three tasks (Figures 3 - 5 in the link).
>
> 3. *The theory and algorithm seem disconnected.*
>
> All components of our paper are built around Abstraction 1. Here’s how they connect:
> **Section 3** focuses on the LDHD challenge and shows that length generalization requires prior knowledge. **Section 4** illustrates how inductive bias can be introduced through model design to address the LDHD challenge. **Section 5** consider practical length generalization techniques from the perspective of Abstraction 1. **Section 5.1** explains the effectiveness of CoT as extending the latent space, which may promote LDHD generalization. **Section 5.2** derive the position embedding design principle according to Abstraction 1, highlighting the importance of handling LDHD generalization in the latent space and the data format separately.
>
> RPE-Square is designed closely following the principle implied by Abstraction 1. The inner "relative distance to some special token" is to handle the unaligned format. The outer "relative distance of relative distance to some special token" is to handle the LDHD generalization in the latent space, which is partially explained by the analysis of PLAA-GRPE. Further details are in Appendix D.1.
>
> 4. *I did not understand Sec. 5.1: how did you determine the latent space of each instance? Why should the latent space take the form in the paper?*
>
> The latent space is determined according to Abstraction 1. We consider the scale $n$ of each instance and a proper $\Sigma$ such that each instance of scale $n$ corresponds to an element in $\Sigma^n$.
>
> With CoT, we treat each step (which requires a model prediction) as an instance of the same scale.
> For example, in the addtion with CoT, the two predition steps, i.e., given $x_0 \dots x_{n-1} + y_0 \dots y_{n-1} =$ to predict $z_0$, and given $x_0 \dots x_{n-1} + y_0 \dots y_{n-1} = z_0$ to predict $z_1$, are treated as separate CoT-step instances of the same scale $n$.
>
> We then decide a proper domain $\bar{\Sigma}$ such that each CoT-step instance of scale corresponds to an element in $\bar{\Sigma}^n$. Furthermore, since CoT typically inserts intermidate results to the original instances, $\bar{\Sigma}$ will take the form $\Sigma\times\Sigma'$. For the addition, $\Sigma'=\lbrace *,0,\dots,9\rbrace$ satisfies the requirement.
>
>
> 5. *...the formulation of LDHD generalization (Def. 1) is quite similar to length generalization... Are there any other applications of this formulation except length generalization?*
>
> Changing "length" to "dimension" is not merely a notion replacement. By the notion "dimension", we highlight the **exponential growth** in sample space as scale increases, and the **complete absence of information** in low-dimension training data about high-dimension behavior. These aspects are not captured by "length", which can be confounded by formatting. Thus, “dimension” more precisely reflects the scaling challenge.
>
> While LDHD generalization is originally to characterize the inherent scaling challenge in length generalization, it could be extended to other applications to character the exponential growth in the sample space size and imperfect information provided by the training data. For example, in graph learning, we may consider the number of nodes as the dimension and use this formulation to describe the generalization from small to large graphs.

---

> > ### Comment · Reviewer_9ohk · 2025-04-03
> >
> > Thank you for your response. I appreciate the following points:
> > - The clarification on the relation between the proposed method and Abstraction 1.
> > - The added experiments.
> >
> > However, I am still not satisfied with the formulation/theory part after reading the rebuttal. As this work is of theoretical nature, I think the following points need to be further discussed:
> > - "Our main contribution is the LDHD generalization formulation (Abstraction 1)...". Note that Abstraction 1 itself is _not_ directly related to LDHD generalization---what it assumes is a _data generation model_ via a latent variable. Similar data generation models are in fact quite common in the representation learning/latent variable modeling literature (e.g., [1,2]). The main difference here is that you explicitly use the generation function $\phi$ to model "data format nuisances". Please note that I am not nitpicking here by saying that Abstraction 1 is not novel. In fact, I agree that considering data format nuisances separately for length generalization is indeed interesting. However, I do have reservations about **whether it is necessary/significant to formulate a new "LDHD generalization problem" based on Abstraction 1 for length generalization** (see below).
> >
> > - You mentioned that "LDHD generalization" differs from conventional length generalization in that it characterizes "the exponential growth in sample space" and "the complete absence of information in low-dimensional training data". Yet, I believe that both aspects are already reflected in length generalization.
> >   - For example, Abbe et al. (2023) [3] discussed the _non-uniqueness of solutions_ when training data is low-dimensional, which implies that the training data lacks information to identify the true concept in the whole sample space, and this is also the main intuition of your no-free-lunch theorem (Theorem 1).
> >   - In Section 5 of [3], the authors note that the length generalization problem can be formulated as "increasing the number of bits" in test data, which naturally corresponds to an _exponential_ increase in the sample space.
> >
> >  - As you have mentioned in your response, the actual difference between "dimension" and "length" is the consideration of _data format confounding_. However, this is already reflected by the data generation model in Abstraction 1, and this generation model could also be integrated into the current length generalization setting. Hence, I do not see why it is necessary to formulate a new learning problem with a new name.
> >
> >  To summarize, from my current angle, the main contents of the paper are two-fold:
> > - Formulating and analyzing a "new" problem of LDHD generalization.
> > - Pointing out that considering data format nuisances is important in length generalization and proposing a method for it.
> >
> > It seems that you mix these two points in the overall presentation, which could be somewhat confusing. In fact, I realized that this is also the main reason why I (and perhaps some other reviewers) found the paper not very easy to read.
> >
> > In my opinion, the second point is interesting and aligns with the proposed algorithm. However, the first point seems to be not valid to me---to my knowledge, the hardness of generalization, the necessity of considering inductive biases, and the actual inductive bias considered (min-degree bias) are all explored in prior work [3], which makes the theoretical part incremental and does not warrant the formulation of a new problem.
> >
> > In this light, I feel it might be necessary for the authors to reconsider the presentation of the paper, which I think seems to require quite a bit of rewriting. Please feel free to correct me if you find any misunderstanding in my comments.
> >
> > ---
> >
> > [1] Hyvärinen and Pajunen. Nonlinear independent component analysis: Existence and uniqueness results. Neural Networks, 1999.
> >
> > [2] Schölkopf et al. Toward causal representation learning. Proceedings of the IEEE, 2021.
> >
> > [3] Abbe et al. Generalization on the unseen, logic reasoning and degree curriculum. ICML, 2023.

---

> > > ### Author Response · Authors · 2025-04-06
> > >
> > > Thank you very much for your further in-depth feedback and constructive suggestions.
> > >
> > > We would like to clarify that our main point is the LDHD generalization perspective/formulation for length generalization, rather than LDHD generalization as a standalone problem. Our motivation is to develop a principled framework that captures the core challenges commonly associated with so-called length generalization.
> > >
> > > To further elaborate, we distinguish three concepts:
> > > - (ideal) length generalization (LG0): some ideal (but unknown) formulation that perfectly captures all the common challenges of generalizing from small-scale instances to large-scale instances;
> > > - length generalization defined on length (LG1): generalization from short sequences to long sequences;
> > > - length generalization from the perspective of LDHD generalization (LG2): our proposal (Abstract 1) to seek LG0.
> > >
> > > In other words, LG0 is the (unknown) "ground truth" notion of length generalization we aim to capture. LG1 and LG2 are two alternative formulations for LG0, through which we can analyze general length generalization problems and design algorithms accordingly.
> > >
> > > 1. *...whether it is necessary/significant to formulate a new "LDHD generalization problem" based on Abstraction 1 for length generalization*
> > >
> > > - *...Abstraction 1 itself is not directly related to LDHD generalization...*
> > >
> > > We respectfully disagree. Abstraction 1 is directly related to LDHD generalization. Specifically, Step 1 (lines 87 - 89) implies length generalization from instances of scale n to those of scale m corresponds to LDHD generalization (in the latent space) from dimension n to dimension m ($n < m$). It is this relation that characterizes the two important properties of LG0: "the exponential growth in sample space" and "the absence of information in training data".
> > >
> > > - *...Similar data generation models are in fact quite common in the representation learning/latent variable modeling literature (e.g., [1,2])...*
> > >
> > > We agree that our abstraction shares technical similarities with prior work in latent variable modeling. However, we believe the significance of any data generation model lies in how it reflects the problem it is intended to characterize. For example, the model in [2] is notable not simply for its technical structure, but for how effectively it captures challenges in causal representation learning. Similarly, while Abstraction 1 resembles other models structurally, its novelty and contribution lie in how well it approximates LG0.
> > >
> > > 2. *...both aspects are already reflected in length generalization.*
> > >
> > > While the previous discussions under the name "length generalization" provide insights and intuitions on which properties LG0 has, to the best of our knowledge, they do not offer a formal, general formulation that explicitly characterizes these properties. For example, while Abbe et al. (2023) [3] mention both increasing "the number of bits" and "the number of -1s", they do not formally explain why or to what extent "both of these variants capture the notion and difficulty of length generalization" in Parity. Also, the failure of length generalization is shown specifically in Parity. In contrast, our proposed formulation (LG2) aims to formalize and unify these insights, serving as an approximation of LG0 that applies beyond individual tasks.
> > >
> > > 3. *...why it is necessary to formulate a new learning problem with a new name*
> > >
> > > We introduce the new name to distinguish LG2 from LG1. The term "length" likely refers to the sequence length and LG1. The term "dimension" emphasizes the two properties that LG0 requires inherently. By explicitly naming our formulation with LDHD generalization, we aim to highlight these conceptual distinctions.
> > >
> > > 4. *It seems that you mix these two points in the overall presentation, which could be somewhat confusing. In fact, I realized that this is also the main reason why I (and perhaps some other reviewers) found the paper not very easy to read.*
> > >
> > > We sincerely appreciate this valuable feedback. We will clarify the two points in the revision. Furthermore, we plan to make the following changes to improve clarity:
> > >
> > > (1) We will explicitly distinguish between the different notions of “length generalization” throughout the paper, e.g., using precise terminology such as LG0, LG1, and LG2 to avoid ambiguity.
> > >
> > > (2) We will make it clear that the central contribution of the paper is the LG2 formulation. Different sections of the paper address different aspects of this formulation, and we will reorganize the presentation to better reflect this structure.
> > >
> > > (3) We will broaden the discussion of related literature, particularly regarding latent variable modeling and previous studies on length generalization. We will clarify the connections to prior work and the novel contributions in our work.

---

### Official Review · Reviewer_osYw · 2025-03-13

**Overall Recommendation:** 4

**Summary:**

Summary: The paper examines Low-Dimension-to-High-Dimension (LDHD) generalization and theoretically demonstrated that LDHD
generalization is unattainable without appropriate inductive bias, focusing on Boolean functions and how different architectures and inductive biases influence this generalization. The study introduces RPE-Square to handle data format nuisances and improve LDHD generalization. Key formulas include the computation of latent variables h and labels y through mappings phi, and the modeling of low-dimension-to-high-dimension generalization with the Boolean function framework.

**Claims And Evidence:**

While the paper presents strong theoretical underpinnings and some experimental results, there are areas where the evidence might not be as convincing or where further clarification and experimentation could strengthen the claims:

1. Generalization of Theoretical Concepts to Practical Applications: The paper does an excellent job in theoretical exposition, especially with the introduction of the No-Free-Lunch Theorem for LDHD generalization. However, the transition from theoretical models to practical, real-world applications is not deeply explored. The claims regarding the practical effectiveness of RPE-Square would be more convincing if supplemented by more diverse and extensive empirical evidence across various domains and tasks.

2. Robustness of RPE-Square: The paper claims that RPE-Square enhances the model's ability to handle data format nuisances effectively. While initial experimental results are provided, the robustness of RPE-Square across a broader range of scenarios and its comparison to other state-of-the-art position embeddings are not thoroughly examined. Claims about its superiority could be better supported by more comprehensive comparative studies that include a variety of datasets and model configurations.

3. Scalability and Performance Across Different Scales: The paper discusses the potential of RPE-Square in managing the challenges associated with LDHD generalization. However, the scalability of this approach—how it performs as the dimensionality of data dramatically increases—is not fully addressed.

**Essential References Not Discussed:**

Not essential but might beneficial:
1. Bellman, R. (1961). Adaptive Control Processes: A Guided Tour. Princeton University Press.
This reference is foundational in the discussion of the curse of dimensionality, providing the initial formal description of the problem which is crucial for understanding the theoretical underpinnings of dimensionality challenges.

2. Tenenbaum, J. B., de Silva, V., & Langford, J. C. (2000). A global geometric framework for nonlinear dimensionality reduction. Science, 290(5500), 2319-2323.
This paper introduces Isomap, a manifold learning technique that provides insights into how high-dimensional data can be effectively reduced, maintaining the intrinsic geometry of the data, which is highly relevant for LDHD generalization.

3. Vincent, P., Larochelle, H., Bengio, Y., & Manzagol, P. A. (2008). Extracting and composing robust features with denoising autoencoders. In Proceedings of the 25th international conference on Machine learning (pp. 1096-1103).
Autoencoders, particularly denoising autoencoders, are useful for dimensionality reduction and feature extraction in noisy datasets, relevant for LDHD generalization where maintaining data integrity across dimensions is critical.

**Experimental Designs Or Analyses:**

In reviewing the paper, I examined the soundness and validity of the experimental designs and analyses, particularly those related to the implementation and evaluation of RPE-Square and its impact on LDHD generalization.

- Data Variability: The experiments would benefit from a more detailed examination of the test data's variability and distribution, particularly in how they mirror real-world scenarios essential for LDHD generalization.
- Computational Load: The paper does not adequately address the computational demands of implementing RPE-Square, crucial for practical applications, especially in resource-constrained environments.

Overall, while the experimental designs and analytical methods employed in the paper are sound for the most part, enhancing the range of benchmarks, incorporating a broader set of evaluation metrics, and providing more details on computational demands and data scales could significantly strengthen the validity of the findings.

**Methods And Evaluation Criteria:**

The proposed methods and evaluation criteria in the paper are appropriate for addressing the challenges LDHD.

Proposed Methods:
- RPE-Square: Enhances traditional Relative Position Embedding by better handling data format nuisances and the shift from low to high-dimensional spaces. This method theoretically supports improved out-of-distribution generalization in deep learning models.
- Chain-of-Thought (CoT): Applies CoT to enhance the understanding of sequences in transformer models, suitable for complex reasoning tasks where sequence understanding is crucial.

Evaluation Criteria:
- Task-Specific Benchmarks: Uses arithmetic reasoning and symbolic logic tasks to evaluate performance, which are appropriate for testing model capabilities in handling length generalization and increased problem complexity.
- Comparison with Baselines: Measures improvements over traditional embeddings and other state-of-the-art methods, providing a clear benchmark for assessing the efficacy of RPE-Square.

Areas for Improvement:
1. Wider Range of Datasets: Including datasets with inherently high-dimensional structures could better test the robustness of the proposed methods.
2. More Diverse Tasks: Expanding evaluations to include varied tasks like image processing could demonstrate the versatility of RPE-Square.
3. Quantitative Metrics: More rigorous metrics that measure performance in high-dimensional settings and assess computational efficiency would provide a comprehensive evaluation.

**Other Comments Or Suggestions:**

Commnets:
- I recommend including "Relative Position Embedding (RPE)" in the abstract for clearer context.
- Figure 1c was unclear until I read lines 117-123 on page 3. Adding a brief explanation beneath the figure could assist readers who seek immediate clarity.
- The first two pages would benefit from a more detailed discussion of goals and applications.

**Other Strengths And Weaknesses:**

Strengths:
It becomes increasingly interesting if you can get through the first two pages!
1. Novel Conceptual Framework: The paper effectively frames LDHD generalization within the broader context of Out-of-Distribution (OOD) generalization, offering a novel perspective on handling high-dimensional data spaces derived from low-dimensional training data.
2. Nice Theoretical Support: The theoretical underpinnings, including the No-Free-Lunch Theorem for LDHD generalization, are robust, providing a strong foundation for the arguments regarding the necessity of inductive biases.
3. Practical Implications: The introduction of RPE-Square as a novel embedding to address both LDHD generalization and data format nuisances is both innovative and practically relevant, offering direct applicability to transformer models.
4. Empirical Validation: The use of Chain-of-Thought (CoT) to demonstrate the practical effectiveness of the proposed methods in restructuring latent space for improved generalization is well-executed, with clear experimental setups and results.

Weaknesses:
As noted earlier, the main areas for improvement include:
1. Enhanced Mathematical Clarity: While foundational proofs like those for the No-Free-Lunch Theorem are well-established, additional mathematical detail could strengthen the theoretical support for RPE-Square.
2. More Empirical Evidence: Broader empirical testing would help validate the practical effectiveness of RPE-Square across various applications.
3. Computational Complexity: Further discussion on the computational demands of implementing RPE-Square is needed to assess its practicality in high-dimensional settings.

**Questions For Authors:**

Questions:
1. Has RPE-Square been evaluated in diverse or challenging environments? What were the challenges? What were the outcomes?
2. Could you provide an insight on the computational overhead of RPE-Square, especially in large-scale or real-time applications?
3. Are there specific conditions or assumptions in Theorem 3 that might limit its general applicability? If so, what are they?

**Relation To Broader Scientific Literature:**

The key contributions of the paper, particularly the development of RPE-Square and its application in LDHD generalization, build on existing scientific literature by enhancing position embedding techniques previously established for transformer models. This development provides a nuanced approach to handling high-dimensional data challenges, extending the utility of relative position embeddings for complex reasoning tasks. The paper's theoretical additions, like the No-Free-Lunch Theorem for LDHD generalization, contribute a rigorous framework to the discussion of dimensionality in machine learning, bridging theoretical concepts with practical machine learning applications. Overall, these contributions effectively integrate and advance the current understanding of embedding designs and generalization in machine learning.

**Theoretical Claims:**

The mathematically oriented proofs, such as those for the No-Free-Lunch Theorem, are rigorous and seem to be correct according to the details provided. However, there's a chance that I might have missed an error. Additionally, the proofs do not thoroughly address the computational complexity of implementing RPE-Square, which is vital for practical applications, particularly in very high-dimensional spaces.

---

> ### Author Rebuttal · Authors · 2025-04-01
>
> We are grateful for your careful evaluation and positive assessment of our work. All experiments results are in the anonymous link: https://www.dropbox.com/scl/fi/52t23nfzev1lo1sq5dmyj/ICML_2025_5857_Rebuttal.pdf?rlkey=5nde6aampze744klvn0rsb3mp&st=37lg3nzo&dl=0.
>
> 1. Has RPE-Square been evaluated in diverse or challenging environments? What were the challenges? What were the outcomes?
>
> We evaluate RPE-Square in three additional tasks: Parity (with CoT), Multiplication (1 * N), and Division (N / 1). RPE-Square achieves length generalization in all the three tasks (Figures 3 - 5 in the link). These results are consistent with our insight of the inductive bias of RPE-Square and further justify the proposed position embedding design principle.
>
> 2. Could you provide an insight on the computational overhead of RPE-Square, especially in large-scale or real-time applications?
>
> Suppose the sequence length is N and the hidden dimension is d. The computational overhead of an attention with RPE-Square is $O(N^4 + N^3 d)=O(N^2 (N^2 + N^d))$ (with the implementation that saves the token attention scores for all query j and key i and reuse them when computing $\text{RPE-Square}_{i,j}$). This overhead can lead to inefficiency for large-scale or real-time applications.
>
> We note that RPE-Square is primarily intended to illustrate the design principle for position embeddings. Our focus is on the challenge of length generalization instead of long-sequence efficiency. We consider it a promising direction to develop more efficient variants of RPE-Square in future work.
>
> 3. Are there specific conditions or assumptions in Theorem 3 that might limit its general applicability? If so, what are they?
>
> Theorem 3 is derived with a simplified model (PLAA-APE). The conclusion in Theorem 3 may not hold precisely for more complex models. Also, the target functions considered in Theorem 3 are restricted to Boolean functions and the notation "min-degree profiler w.r.t. linearly independent set" may not be applied to non-Boolean functions directly. Additionally, the result include a limit ($\hat{P}$) but we cannot use the exact limit in practice.
>
> Despite the above theoretical limitations, we think Theorem 3 can provide sufficient insight on the inductive bias of APE. The simplified model captures the essential effect of APE on the inductive bias. The analysis in the Boolean functions can be naturally extended to functions defined on finite domains. The limit can be well approximately by choosing a sufficient small initialization.
>
>
> We also appreciate your suggestions on writing to improve clarity and readability. We will revise our manuscript accordingly.

---

### Official Review · Reviewer_H9fc · 2025-03-14

**Overall Recommendation:** 2

**Summary:**

This paper introduces the concept of Low-Dimension-to-High-Dimension (LDHD) Generalization to address challenges in out-of-distribution (OOD) generalization, particularly in reasoning tasks where models are trained on low-dimensional subspaces and tested on higher-dimensional spaces. The authors propose that LDHD generalization is fundamental to understanding length generalization, where models extend learned patterns to longer sequences.

**Claims And Evidence:**

The paper presents several key claims:
1. LDHD Generalization Necessitates Inductive Biases: The authors assert that without appropriate inductive biases, models cannot generalize from low-dimensional training data to high-dimensional testing scenarios. This claim is supported by the No-Free-Lunch Theorem for LDHD generalization, which mathematically establishes the necessity of inductive biases.

2. CoT Enhances Length Generalization: The paper suggests that Chain-of-Thought prompting improves length generalization by restructuring the latent space, facilitating better LDHD generalization. While the theoretical analysis supports this claim, direct empirical comparisons of models trained with and without CoT on reasoning tasks are lacking.

3. RPE-Square Improves Length Generalization: The authors propose that RPE-Square, their novel position embedding method, addresses both LDHD generalization and data format inconsistencies, leading to improved length generalization. Experimental results on addition and unaligned copy tasks demonstrate its effectiveness. However, comparisons with alternative position embeddings designed for length generalization, such as ALiBi, RoPE, NoPE, Abacus, as well as position coupling are not included.

Overall, the claims are theoretically substantiated, but additional empirical validation on diverse tasks would strengthen the conclusions.

**Essential References Not Discussed:**

I am not aware of essential references not discussed.

**Experimental Designs Or Analyses:**

The experiments are well-structured but exhibit certain limitations:

- Lack of Generalization Tests on Longer Sequences: It remains unclear whether models trained with RPE-Square can extrapolate to significantly longer sequences than those encountered during training.
- Absence of Robustness Analysis: The performance of RPE-Square across various tasks beyond addition and unaligned copy is not explored.
- Experiments comparing with and without CoT seems to be missing.

**Methods And Evaluation Criteria:**

Theoretical analysis is sound and clearly presented. The evaluation tasks (Unaligned Copy, URF Addition) are appropriate for illustrating their claims. However, broader validation on more varied or real-world data could strengthen applicability claims.

**Other Comments Or Suggestions:**

- typo: line 168 overparatermization -> overparameterization
- typo: line 202 equation X_m instead of X^m in the subscript of E
- discuss briefly computational complexity of RPE-Square in practice.
- the definition of "concept c" in Abstraction 1 seems missing.

**Other Strengths And Weaknesses:**

### Strengths:
- Novel Formalization: The introduction of LDHD generalization offers a valuable perspective on length generalization challenges.
- Practical Contribution: The development of RPE-Square presents a tangible improvement for length generalization tasks.

### Weakness:
- I find it hard to make connections between sections and get a clear intuition behind the approach.
- Limited diversity of tasks considered: The paper only considers two simple tasks of unaligned copy and URF addition of small scale.
- Lacks comparison with diverse Positional Embeddings, developed to tackle length generalization, leaving the relative effectiveness of RPE-Square unclear
- Computational overhead of RPE-Square not discussed.

**Questions For Authors:**

- Generalization of theoretical results: Do you have empirical evidence confirming that actual Transformers trained in realistic scenarios indeed favor minimum-degree interpolants?
- Computational Complexity: What is the computational and memory overhead of RPE-Square compared to standard positional encoding, especially for longer sequences?

**Relation To Broader Scientific Literature:**

The paper appropriately references key works on:
- Length Generalization in Transformers: Citing studies by Anil et al. (2022) and Jelassi et al. (2023).
- Inductive Bias in Neural Networks: Expands on works by Gunasekar et al. (2017), Abbe et al. (2023), providing a clear link to length generalization.

**Theoretical Claims:**

Yes, I checked Theorem 1 (No Free Lunch Theorem) and the proof in the supplementary material seems correct.

---

> ### Author Rebuttal · Authors · 2025-04-01
>
> Thank you for your time and valuable feedback. We'd like to response to your concerns as below.
> All experiments results are in the anonymous link: https://www.dropbox.com/scl/fi/52t23nfzev1lo1sq5dmyj/ICML_2025_5857_Rebuttal.pdf?rlkey=5nde6aampze744klvn0rsb3mp&st=37lg3nzo&dl=0.
>
> 1. *...comparisons with alternative position embeddings designed for length generalization...*
>
> We conduct additional experiments to compare the mentioned position embeddings (Since Abacus and position coupling are similar apart from some subtle implementation details in our tasks, we only implement Abacus currently.) The results (Figures 1 - 2 in the link) show that RPE-Square achieves the best overall length generalization in the tasks.
>
> 2. *Lack of Generalization Tests on Longer Sequences*
>
> We evaluate the models with RPE-Square on significantly longer sequences in three tasks (Parity (with CoT), Multiplication (1 * N), and Division (N * 1)) The results (Figure 7 in the link) show that the models can extrapolate to significantly longer sequences to some extend and the performance may decreases as the sequence length increases.
>
> 3. *Absence of Robustness Analysis: The performance of RPE-Square across various tasks beyond addition and unaligned copy is not explored.
>
> We apply RPE-Square to three new tasks: Parity (With CoT), Mulplication (1 * N), and Division (N / 1). The results (Figures 3 - 5 in the link) show that Transformers equipped with RPE-Square can achieve length generalization in all the three tasks, showing the robustness of RPE-Square across different tasks.
>
> 4. *Experiments comparing with and without CoT seems to be missing.*
>
> We compare the length generalization of Parity with CoT and without CoT. We train Transformers with RPE-Square. We only achieve length generalization for Parity with CoT (Figure 8 in the link). This is because as CoT enables LDHD generalization via relative distances in the latent space. There are also many empirical results showing CoT can enhance length generalization in previous works, e.g., [1] [2].
>
> [1] Zhou, H., et al., 2023. What algorithms can transformers learn? a study in length generalization. arXiv preprint arXiv:2310.16028.
>
> [2] Feng, G., et al., 2023. Towards revealing the mystery behind chain of thought: a theoretical perspective. NeurIPS.
>
> 5. *I find it hard to make connections between sections and get a clear intuition behind the approach.*
>
> (a) Connections between sections.
>
> **Section 1** introduces the problem of length generalization and proposes LDHD generalization to address the mismatch between input length and problem scale. **Based on the LDHD formulation**,  **Section 3** presents the No-Free-Lunch Theorem for LDHD generalization, motivating the need for inductive bias. **Section 4** consider how different inductive bias can be incorporated by choosing different models, a technical problem to address the LDHD generalization challenge in Section 3. In **Section 5**, we further discuss the implications of the LDHD generalization formulation for practical length generalization. We explain why CoT can promote length generalization. We also propose the position embeddding design principle for length generalization: consider the inherent LDHD generalization and the data format nuisance separately.
>
> (b) Intuition behind the approach.
>
> - The intuition behind LDHD generalization formulation includes: (1) The sample space grows exponentially as the scale increases and the small-scale instances do not tell how large-scale instances can be solved without external information; (2) In language modeling, the sequence length can be affected by the data format and may not faithfully reflect the problem scale.
>
> - The intuition behind RPE-Square: the unaligned formats can be handled by the relative distances to some special tokens. Detailed illustrations are in Appendix D.1.
>
> 6. *the definition of "concept c" in Abstraction 1 seems missing.*
>
> A "concept" refers to a target mapping, as in [1]. We'll clarify this in the revision.
>
> [1] Mohri, Mehryar, 2018. Foundations of machine learning, Chapter 2, The PAC Learning Framework, Section 2.1, The PAC learning model.
>
> 5. *Generalization of theoretical results*
>
> We provide empirical support (Figure 9) by training GPT2 with APE and RPE on different functions. Probing reveals the learned models roughly align with minimum-degree interpolators w.r.t. their respective bases.
>
> 6. *... the computational and memory overhead ...*
>
> Suppose the sequence length is N and the hidden dimension is d.
> RPE-Square: $O(N^4 + N^3 d)$ compute and $O(N^2 + N d)$ memory.
>
> Standard PE: $O(N^2 d)$ compute and $O(N^2 + N d)$ memory.
>
> While RPE-Square is less efficient than the standard position embedding, it mainly serves as a proof-of-concept for separating LDHD and format handling. It is focused on the length generalization challenge instead of the challenge of long sequence. Designing efficient variants is a promising direction for future work.

---

### Official Review · Reviewer_4S48 · 2025-03-16

**Overall Recommendation:** 3

**Summary:**

*Despite my best efforts, I found this paper very hard to parse. I am discounting my confidence to reflect the same.*

The paper studies Low-Dimension to High-Dimension generalization (LDHD) problem, a special case of OOD generalization. At its core, the paper argues the impossibility of generalizing to high-dimensions without imposing inductive biases or prior knowledge. They then consider implications of their argument on length generalization for unaligned copy and addition tasks, and propose RPE-Square (relative position embedding) that leverages some domain knowledge such as the distance to the special tokens (such as =, +, [BoS], [EoS]). The paper then demonstrated RPE-Square's generalization to high-dimensions.

The main contributions of the paper are the following. (a) a formalization to disentangle the data format and dimension of the problem, (b) through analysis on binary functions and various positional encodings (relative, absolute and general), the contribution to the attention scores from positional embedding is min-degree interpolator wrt to a linearly independent set, (c) a new positional embedding scheme: RPE-Square.

**Claims And Evidence:**

The paper's analysis (see (b) from my paper's contribution list above) makes the following choices: (1) isolate and analyse only the positional embedding contribution to the attention score, (b) assume the value head is an identity, (c) restricts only to binary functions.
However, the paper did not justify how their choices do not compromise the generality of their analysis.

Their proposed positional embedding scheme: RPE-Square relies on domain knowledge of the task in hand. It is expected that the paper at least present how their approach extends to other tasks, and what kind of domain knowledge can be exploited for different problems.

Regarding OOD and LDHD. I believe the paper's analysis do not explain why we observe (surprising) LDHD generalization for several other problems without any special domain knowledge. For example, consider the multilingual abilities of LLMs despite <10% representation of non-english data in pretraining. Given the typically low language representation, it is possible that many (entity, language) combinations are not present in the training data, yet the models demonstrated generalization to such unseen combinations (see [1] for an example). We may see this as LDHD generalization where (entity, language) defines a dimension.

[1] Chen, Yang, Yitao Liang, and Zhouchen Lin. "Low-Dimension-to-High-Dimension Generalization And Its Implications for Length Generalization." arXiv preprint arXiv:2410.08898 (2024).

**Essential References Not Discussed:**

NA, please see my response to the next question.

**Experimental Designs Or Analyses:**

Yes, their only experiment looks sound.

**Methods And Evaluation Criteria:**

Yes, the paper pertains LDHD generalization and length generalization is a good task.
The only evaluation the paper conducted is with addition and copy tasks. The choice of tasks make sense.

**Other Comments Or Suggestions:**

L92, second column, formant-> format.
L221, first column. I believe it is $$n=\arg\min_k \\{h\in \Sigma^k\\\}$$

**Other Strengths And Weaknesses:**

**Strengths**
LDHD generalization is a compelling problem, and studying the SoTA models for LDHD is even more.
The analysis, although on a special class of functions and assumptions, present some interesting insights into generalization.

**Weakness**
The writing of the paper needs much improvement. They should aim to reduce the number of notations introduced, explain in text every equation and symbol before presenting the math expression. Present proof sketches in the main paper. Make a more coherent and less dense read by taking care of the reasoning-related sections and Free-lunch thm section.

**Questions For Authors:**

Please address my questions listed under "Claims and Evidence".

**Relation To Broader Scientific Literature:**

The findings need not surprise a significant class of researchers working on causality or symbolic methods. But the folks who bet on learning purely from signal, large scale training and exploration would point the paper to many phenomena that the paper do not explain like the way I did in the last point of the Claims section.

**Theoretical Claims:**

I did not check the correctness of any theorem, and only glanced at the statements.

---

> ### Author Rebuttal · Authors · 2025-04-01
>
> Thank you for your thoughtful and constructive feedback. Below, we respond to your concerns.
> All experiments results are in the anonymous link: https://www.dropbox.com/scl/fi/52t23nfzev1lo1sq5dmyj/ICML_2025_5857_Rebuttal.pdf?rlkey=5nde6aampze744klvn0rsb3mp&st=37lg3nzo&dl=0.
>
> 1. *... However, the paper did not justify how their choices do not compromise the generality of their analysis.*
>
> (a) We isolate the positional contribution in attention to highlight how position embeddings affect inductive bias and length generalization. This helps us better understand their standalone impact, and the empirical results support this analysis.
>
> (b) We assume an identity value head because it is not central to the question of length generalization. Intuitively, we expect that the interpolator learns the correct value head to perform in-distribution generalization. So when focusing on length generalization, it is reasonable to assume a correct value head and analyze how the model handles larger-scale instances. This simplification helps clarify the role of attention and position embeddings.
>
> (c) While our analysis focuses on binary functions, the core insights naturally extend to tasks over finite alphabets, because we can consider the binary representations. For example, for an alphabet of size $N$, each symbol can be encoded using $log N$ bits, and the analysis for binary functions can be applied similarly. Though not identical, the notion of the "minimum-degree interpolator over a linearly independent set" continues to hold in spirit under binary encoding schemes. Thus, the results we derive for binary settings are relevant to broader cases like language modeling.
>
> Thank you for pointing out this oversight and we will include the justification in our manuscript.
>
> 2. *... It is expected that the paper at least present how their approach extends to other tasks, and what kind of domain knowledge can be exploited for different problems.*
>
> We conducted additional experiments to address this concern:
>
> (a) We apply RPE-Square to three new tasks: Parity (With CoT), Mulplication (1 * N), and Division (N / 1), and compared its performance against more baselines (RPE, RoPE, NoPE, ALiBi, Abacus). The results (Figures 3 - 5 in the link) show that RPE-Square can also achieve length generalization in these tasks, demonstrating its applicability beyond the initial two tasks.
>
> (b) We also consider a task called AdditionMod10, which requires different domain knowledge from what RPE-Square captures. AdditionMod10 computes the sum of the addends modulo 10. The result depends only on the first digits of the addends (due to the modulo operation), so positional embedding needs to capture the absolute value of the relative distances to some special tokens ([BOS] and "+"). Following the same design principle, we consider a new position embedding called RPE-Absolute to encode "the absolute value of the relative distances to some special tokens". The experiment result (Figure 6 in the link) shows that RPE-Absolute achieves length generalization in AdditionMod10. This suggests that our position embedding design approach is adaptable to different domain knowledge.
>
> 3. *Regarding OOD and LDHD. I believe the paper's analysis do not explain why we observe (surprising) LDHD generalization for several other problems without any special domain knowledge...*
>
> (a) Domain knowledge can be implicitly encoded via architecture design, training algorithms, data distributions, preprocessing strategies, etc. Even if domain knowledge is not injected explicitly, these prior choices reflect inductive biases tailored to the task.
>
> (b) "...despite <10% representation of non-english data in pretraining"
>
> It is important to distinguish low representation from zero representation. Prior work [1] shows that adding even a very small amount of long sequences to the training set can dramatically improve length generalization, which is called priming. From our perspective, this is because the presence of long sequences, even in small quantities, prevents the model from learning an overly simple interpolator that fails to extrapolate. For example, in digit-wise addition, if the training set only includes 3-digit numbers, the model might learn to ignore digits beyond the third. However, including a few 5-digit samples forces the model to also consider later digits, improving its extrapolation ability.
>
> (c) While this work is mainly focused on length generalization, it is an interesting future direction to study how the compositional generalization can be understood from the perspective of LDHD generalization.
>
> [1] Jelassi, S. et al., 2023. Length generalization in arithmetic transformers. arXiv preprint arXiv:2306.15400.
>
> 4. *The writing of the paper needs much improvement...*
>
> We will carefully proofread our submission in the revision.

---

> > ### Comment · Reviewer_4S48 · 2025-04-05
> >
> > Thanks for the response. My concerns are well-addressed. I suggest the authors mention the point about generality and differences in zero vs low representations in the revised paper.  I am updating my score accordingly.

---

> > > ### Author Response · Authors · 2025-04-07
> > >
> > > Thank you very much for reconsidering the score and for the helpful suggestions. As recommended, we will discuss the point regarding generality and the distinctions between zero and low representations in the revised manuscript.

---

### Decision · Program_Chairs · 2025-05-01

**Decision:**

Accept (poster)

**Comment:**

This paper studies Low-Dimension-to-High-Dimension (LDHD) generalization for Boolean functions using different architectures, under inductive bias. Based on the insights, the authors propose a position encoding method, called RPE-Square to facilitate LDHD generalization. They perform experiments on LLMs with transformer architectures and show promising results on simple Math tasks.

The reviewers have raised two main concerns -- lack of readability and limited experimental evaluation. The authors have addressed both these issues by providing explanations in the rebuttal (some reviewers have acknowledged that they now have a better understanding of the results) and providing additional experimental results. If these improvements are included in the final version of the paper, the paper makes solid contributions.